# IL-17A-producing NKp44(−) group 3 innate lymphoid cells accumulate in Familial Adenomatous Polyposis duodenal tissue

Kim M. Kaiser[1], Jan Raabe[1], Michael ToVinh[1], Gudrun Hack[1], Sarah Ahmad[1], Niko Müller[1], Julia Cassella[1], Sofia I. Walravens[1], Paula Alfaro[1], Lauren Arias Garcia[1], Dominik J. Kaczmarek[1,2], Tim Marwitz[1,2], Felix Goeser[1], Hans Dieter Nischalke[1], Philipp Lutz[1], Nils Sommer[3], Tim Vilz[2,3], Marieta Toma[4], Susanne Steiner[4], Oliver Hommerding[2,4], Johannes Oldenburg[5], Michael Hölzel[6], Sebastian Kadzik[6], Alexander Maas[7], Jonas Eckrich[8], Philipp Zumfelde[9], Farhad Shakeri[10], Svetozar Nesic[10], Andreas Buness[11], Emilia De Caro[12,13], Matthias Becker[12], Marc D. Beyer[12,14,15], Thomas Ulas[12,13,15], Anna C. Aschenbrenner[12], Lisa M. Steinheuer[6], Kevin Thurley[6,16], Sandy Kroh[16,17], Ralf Uecker[17,18], Anja E. Hauser[17,18], Florian N. Gohr[19], Florian I. Schmidt[19], Danni Wang[20], Kathrin Held[20,21], Olga Baranov[20], Christof Geldmacher[20,21], Christian P. Strassburg[1,2], Robert Hüneburg[1,2,22], Benjamin Krämer[1,22] ✉ & Jacob Nattermann[1,2,21,22] ✉

Familial adenomatous polyposis (FAP) is an inherited gastrointestinal syndrome associated with duodenal adenoma formation. Even among carriers of the same genetic variant, duodenal phenotypes vary, indicating that additional factors, such as the local immune system, play a role. We observe an increase in duodenal IL-17A(+)NKp44(−) innate lymphoid type 3 cell (ILC3) in FAP, localized near the epithelium and enriched in adenomas and carcinomas. Elevated *IL1B*, *IL23A*, and *DLL4* transcript levels correlate with IL-17A(+)NKp44(−)ILC3 accumulation, and in vitro studies with duodenal organoids confirmed this relationship. Bulk RNA sequencing reveals upregulated Reactive oxygen species (ROS)-inducing enzymes *DUOX2* and *DUOXA2* in FAP adenomas. IL-17A-stimulated FAP organoids show increased *DUOX2/DUOXA2* expression, Duox2 protein, and ROS production, leading to DNA damage, suggesting a mechanism by which these immune cells promote tumorigenesis. These findings suggest IL-17A(+)NKp44(−)ILC3s may contribute to a local environment that makes the epithelium more submissive for oncogenic transformation in FAP.

Familial adenomatous polyposis (FAP), an autosomal dominant inherited gastrointestinal tumor syndrome caused by a pathogenic germline mutation in the adenomatous polyposis coli (*APC*) gene, is characterized by the development of a multitude (100–1000) of colorectal adenomas[1–3]. Without prophylactic colectomy FAP patients will almost inevitably develop colorectal cancer (CRC). Besides CRC, FAP is also associated with a variety of extracolonic manifestations[4,5], with duodenal cancers representing a major cause of mortality in FAP patients after colectomy[6].

---

Apart from colonic polyposis and CRC, the occurrence of duodenal adenomas is the most common intestinal manifestation of FAP. Currently available data indicate the lifetime risk of developing duodenal adenomas to be nearly 100%, with an estimated lifetime risk for occurrence of duodenal carcinoma of 4–12%[7,8], which is significantly higher than in the general population (<1%)[9].

Based on the fact that the risk for carcinoma development is highest in patients with severe duodenal polyposis[10] and the observation that adenoma tissue either as a component of or in close proximity to duodenal carcinoma has been found in over 90% of malignancies[11], duodenal cancer is thought to develop from duodenal adenomas. This supports the idea of the adenoma-carcinoma sequence in the duodenum resembling that in CRC.

However, in contrast to CRC, only a proportion of FAP patients develop duodenal cancer and the extent of duodenal polyposis varies considerably. Even between carriers of the same genetic variant in the same family, duodenal phenotype and clinical courses vary, indicating that, in addition to the genotype, other factors play a role[8,12].

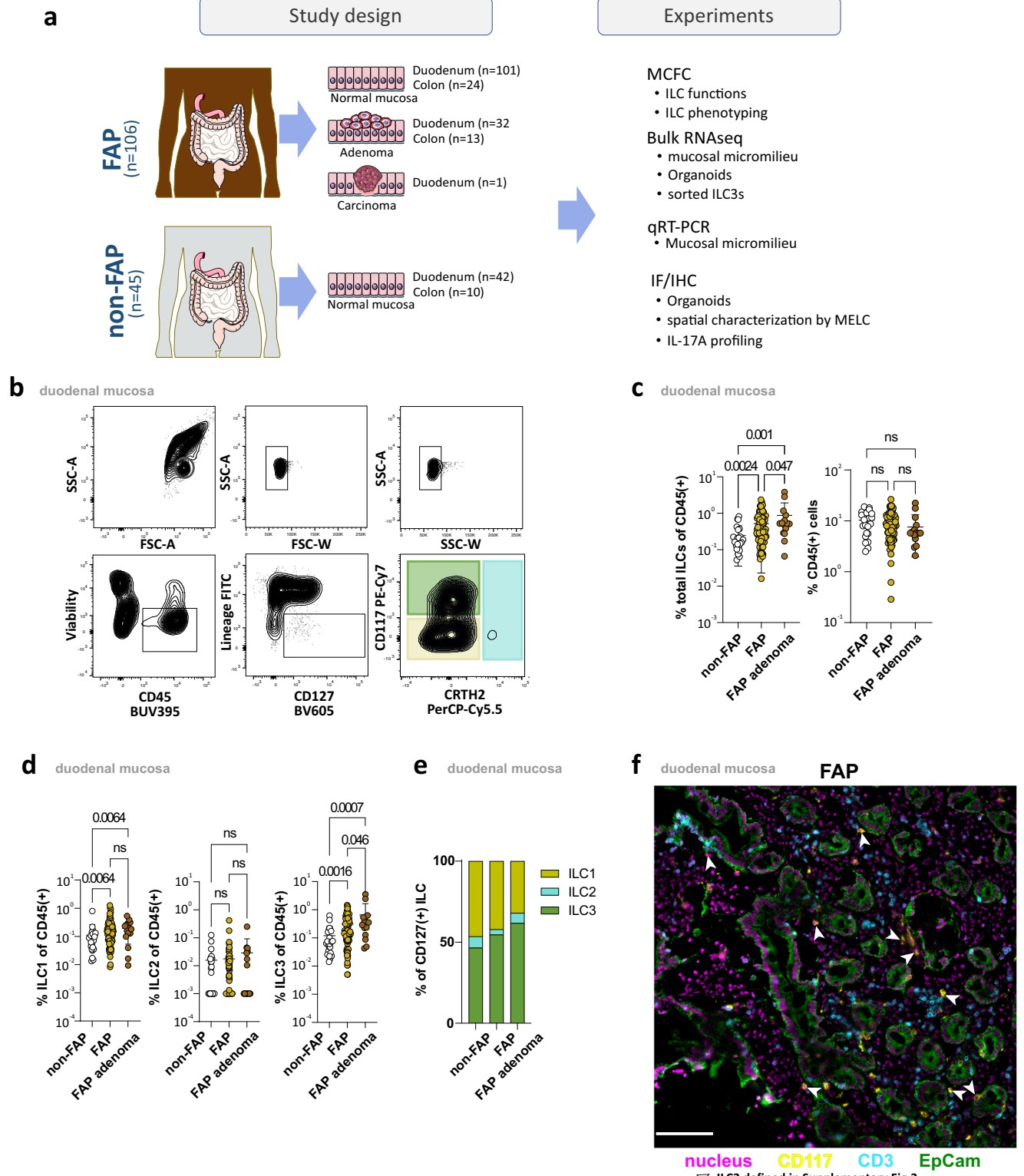

**Fig. 1 | Increased frequencies of CD127( + )CD117( + )ILC3 in FAP duodenal tissue.**
**a** Study design. This figure includes images from Servier Medical Art (https://smart.
servier.com/category/anatomy-and-the-human-body/digestive-system/ & https://
smart.servier.com/smart_image/simple-columnar-epithelium), licensed under a
Creative Commons Attribution 4.0 Unported License (https://creativecommons.
org/licenses/by/4.0). The images were modified to depict tumor tissue within the
epithelium. **b** Representative FACS plots showing gating strategy from duodenal
lymphocytes to ILC1s (yellow), ILC2s (blue), and ILC3s (green). Lineage for this
staining is defined as CD3, CD4, CD5, CD14, CD19, CD20, TCRγδ, TCRαβ, BDCA-2,
CD1a, CD34, NKp80, CD94, FcεR1a and CD123. **c** Proportion of total Lin(−)CD127(+)
ILCs among CD45(+) cells in duodenal adenomatous ($n = 15$) and normal ($n = 76$)
duodenal tissue of FAP patients, as well as normal duodenal mucosa ($n = 24$) of non-
FAP controls (left panel). Frequencies of CD45(+) cells in duodenal adenomatous
($n = 15$) and normal mucosa ($n = 76$) of FAP patients, and normal mucosa ($n = 24$) of
non-FAP controls (right panel). Mean ± SD. **d** Proportion of total Lin(−)CD127( + )
ILC1, ILC2, and ILC3 among CD45(+) cells in duodenal adenomatous ($n = 15$) and
normal ($n = 76$) duodenal tissue of FAP patients, as well as normal duodenal mucosa
($n = 24$) of non-FAP controls. Mean ± SD. **e** Proportion of ILC subsets ILC1(yellow),
ILC2(blue), and ILC3(green) among Lin(−)CD127(+) cells in duodenal adenomatous
($n = 15$) and normal ($n = 76$) tissue of FAP patients, and normal mucosa ($n = 24$) of
non-FAP controls. **f** Representative MELC images of FAP normal tissue showing
ILC3 cells, indicated by white arrows, with CD117(yellow), CD3(blue), EpCAM(-
green), and nucleus stainings(violet). ILC3 cells are defined in Supplementary Fig. 3
as CD45( + )CD117( + )CD3(−)CD14(−)CD16(−)CD19(−)EpCAM(−) lymphoid cells.
The white scale bar represents 100 μm. Based on three independent biological
replicates. Statistical significance analyzed by the Kruskal−Wallis (KW) test cor-
rected for multiple comparisons using FDR (Benjamini, Krieger, Yekutieli). *q*-values
are indicated; ns = not significant. Non-FAP is white, normal FAP is ochre and FAP
adenomas are brown in the c&d subdivisions diagrams.

The local immune system is of special interest in this context as
many studies confirmed the effect of local immune responses on the
development, progression, and treatment outcome in a variety of dif-
ferent tumors[6]. Among the cells with a proposed role in modulating
tumor formation are innate lymphoid cells (ILCs), a subset of tissue-
resident lymphocytes lacking T-cell-specific receptors. ILCs encompass
IFN-γ and TNF-α-producing group 1 ILC (ILC1), IL-5 and IL-13-secreting
ILC2 and ILC3, which are capable of producing IL-17A or IL-22[13–15].

Beyond their role in mediating immunity against pathogens and
maintaining tissue homeostasis at mucosal sites[16–18] emerging data
indicate ILCs to play an important role in tumor formation and
course[19,20]. Regarding gastrointestinal tumors, however, the exact
involvement of ILCs is currently incompletely understood and data on
the role of ILCs in duodenal tumorigenesis are particularly scarce.

In this work, we analyze duodenal tissue from a large cohort of FAP
patients and present initial evidence that IL-17A-producing ILC3s may
shape a duodenal microenvironment conducive to oncogenic trans-
formation. Targeting these cells may, therefore, represent a promising
therapeutic approach to prevent the formation of duodenal carcinoma.

## Results
### Accumulation of CD127(+)CD117(+)ILC3s in FAP duodenal tissue
To assess the role of ILCs in duodenal polyposis, we analyzed normal,
adenomatous, and carcinoma tissue samples obtained from 106 FAP
patients and 45 non-FAP controls. Tissue-resident lymphocytes were
analyzed using multicolor flow cytometry, IF/IHC, and tested for their
interaction with duodenal organoids. The duodenal microenviron-
ment was analyzed by qRT-PCR, spatial microscopy and bulk RNAseq.
Colon tissue (FAP $n = 24$, non-FAP $n = 10$) was studied as additional
control (Fig. 1a and Supplementary Data 1).

ILCs were characterized as being CD45( + )CD127(+)Lin(−) (CD3,
CD4, CD5, CD14, CD19, CD20, TCRγδ, TCRαβ, BDCA-2, CD1a, CD34,
NKp80, CD94, FcεR1a, and CD123) and further classified as ILC1, ILC2,
and ILC3 based on the expression of CD117 and CRTH2[13,14] (Fig. 1b and
Supplementary Fig. 1a). Using this approach, we found frequency of
total duodenal ILCs (CD45( + )CD127(+)Lin(−)) to be significantly
increased in FAP patients compared to non-FAP controls. Even in
macroscopically normal duodenal mucosa, significantly increased ILC
frequencies were observed in FAP patients, with the highest number
observed in adenomatous tissue samples (Fig. 1c). No such differences
were found regarding the frequency of total CD45(+) lymphocytes,
indicating specificity of our finding (Fig. 1c). A more detailed analysis of
the duodenal ILC compartment demonstrated CD117(−)CRTH2(−)ILC1s
and CD117( + )CRTH2(−)ILC3s to represent the major duodenal ILC
subsets, whereas CRTH2(+)ILC2s were barely detectable in the duode-
nal mucosa of both FAP patients and non-FAP controls (Fig. 1d, e and
Supplementary Fig. 1b). Compared to non-FAP controls, frequencies of
duodenal ILC1s were significantly elevated in FAP patients (Fig. 1d). The
most striking alterations, however, were found for the ILC3 subset.

Here, we not only observed significantly increased frequencies in nor-
mal and adenomatous FAP mucosa compared to controls but also sig-
nificant differences between non-adenoma and adenoma tissue in FAP
patients (Fig. 1d). Accordingly, ILC3s represented the dominant ILC
subset in FAP adenomas, suggesting these cells to play a prominent role
in FAP-associated oncogenic transformation (Fig. 1e).

Multi-Epitope-Ligand-Cartography (MELC) analyses demon-
strated ILC3s to be primarily located in the epithelial area with a pre-
dominant presence in the subepithelial region, as shown by EPCAM co-
staining (Fig. 1f).

### Elevated IL-17A production of duodenal NKp44(−)ILC3s in FAP
Next, we analyzed ILC3s for their capacity to produce cytokines con-
sidered characteristic of ILC3s by flow cytometry[21]. Duodenal ILC3s
exhibited only negligible production of IL-22 but were characterized
by robust production of IL-8, TNF-a, and IL-2, respectively. However,
no differences were found between FAP and non-FAP controls (Sup-
plementary Fig. 2a). In contrast, we observed frequencies of IL-17A-
producing ILC3s to be significantly increased in FAP mucosa compared
to non-FAP controls. This increase in IL-17A-producing ILC3s was
already observed in macroscopically normal mucosa in FAP but was
most prominent in adenomatous tissue. Analyzing the proportion of
IL-17A(+)ILC3s relative to total CD45(+) lymphocytes (Fig. 2b), con-
firmed a FAP-associated increase in IL-17A production by ILC3s.

Consistent with previous reports[22,23], NKp44(−)ILC3s were identi-
fied as the primary IL-17A-producing ILC3 subset (Fig. 2c), specifically
accounting for the elevated IL-17A production in FAP (Fig. 2d, e).

A phenotypic comparison between NKp44(−) and NKp44(+) ILC3s
showed that the NKp44(−) subset had fewer CD56-expressing cells−
with a significant difference observed only in the non-FAP control
group− while no differences were found in other commonly used
ILC3 markers (Supplementary Fig. 4b, c). Furthermore, bulk RNA-
sequencing of FAP adenomas revealed that NKp44(−) ILC3s upregu-
lated several immune-related pathways compared to their NKp44(+)
counterparts, including "immune receptor activity," "chemokine sig-
naling," and "cell adhesion molecules" (Fig. 2f). These findings suggest
that NKp44(−) ILC3s may play a crucial role in local immune responses
and interactions within the microenvironment.

To determine the spatial localization of IL-17A-producing ILC3s
within the tissue, we employed immunohistochemistry (IHC). Our
analysis revealed that the vast majority of IL-17A+ cells lacked CD3
expression, suggesting that Th17 cells play a minor role in this context
(Fig. 3a). Consistent with these findings, we observed that - unlike ILC3s -
there were no differences between the groups in the frequency of IL-
17A-producing duodenal CD4( + ) T cells after PMA stimulation (Fig. 3b).

IHC analysis revealed IL-17A( + )CD3(−) cells to be predominantly
located in the epithelial area. This finding was further corroborated by
MELC. Although direct detection of IL-17A was not feasible, we
employed phenotypic markers to identify and localize NKp44(−)ILC3,

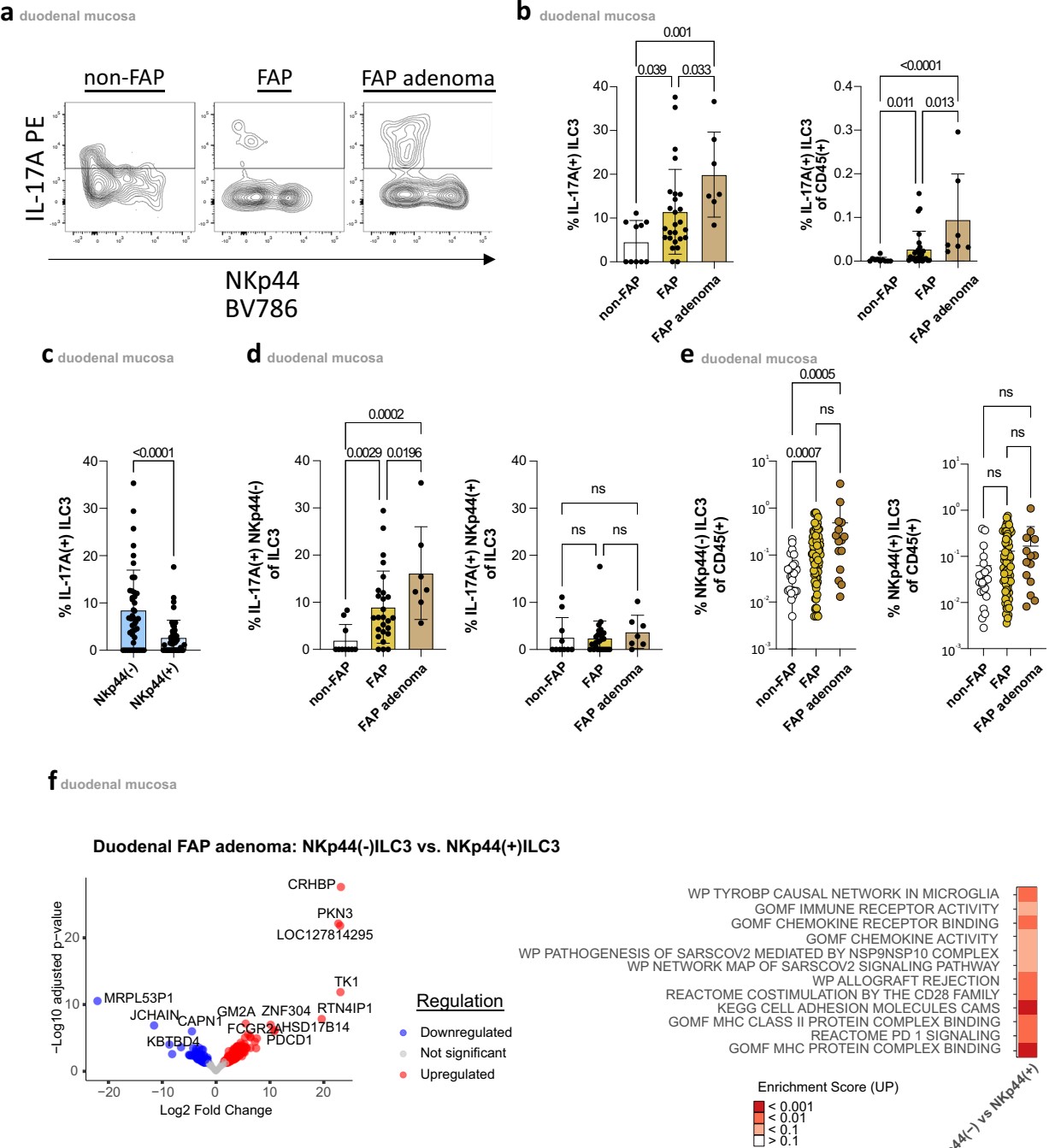

**Fig. 2 | Increased frequencies of IL-17A-producing NKp44(−)ILC3 in FAP duodenal tissue. a** Representative dot blot displaying intracellular IL-17A production of ILC3 in adenomatous and normal mucosa of FAP compared to normal mucosa of non-FAP controls following PMA/Ionomycin stimulation. **b** Percentages of IL-17A( + )ILC3 in duodenal adenomatous (*n* = 7) and normal mucosa (*n* = 26) of FAP patients, and normal mucosa (*n* = 10) of non-FAP controls following PMA/Iono-mycin stimulation (left panel). Gating strategy of respective subsets in Supplementary Fig. 9a. Mean ± SD. Frequency of IL-17A( + )ILC3 among CD45(+) cells in duodenal adenomatous (*n* = 7) and normal (*n* = 26) mucosa of FAP patients and normal mucosa (*n* = 10) of non-FAP controls following PMA/Ionomycin stimulation (right panel). Gating strategy of respective subsets is in Supplementary Fig. 9a. **c** Percentages of IL-17A(+)cells among duodenal NKp44(+)(blue)(from 43 donors) and NKp44(−)ILC3(light blue)(from 43 donors) following PMA/Ionomycin stimulation. Gating strategy of respective subsets is in Supplementary Fig. 9a. **d** Proportions of IL-17A(+)cells within NKp44(+) and NKp44(−)ILC3 in duodenal adenomatous (*n* = 7) and normal mucosa (*n* = 26) of FAP patients, and normal mucosa (*n* = 10) of non-FAP controls following PMA/Ionomycin stimulation. Gating

strategy of respective subsets is in Supplementary Fig. 9a. **e** Frequency of NKp44(−) ILC3 among CD45(+) cells in duodenal adenomatous (*n* = 15) and normal mucosa (*n* = 76) of FAP patients and normal mucosa (*n* = 24) of non-FAP controls. Gating strategy of respective subsets in Supplementary Fig. 9a. Mean ± SD. **f** Bulk RNA-seq analysis showing differentially expressed genes (DEGs) (up- in red and down-regulated genes in blue) in a volcano plot with adjusted *p*-value and Log2 fold-change (left panel), with the corresponding enrichment analysis indicating sig-nificant pathways based on the enrichment score (white-red scale)(right panel). Sorting strategy for respective subsets in Supplementary Fig. 9c. Statistical sig-nificance analyzed by Two-tailed Wilcoxon matched-pairs signed rank test (**c**), Kruskal–Wallis (KW) test (**b**, **d**, **e**) corrected for multiple comparisons using FDR (Benjamini, Krieger, Yekutieli). q(KW test)- and p(Wilcoxon)-values are indicated; ns = not significant. DEGs (**f**) were analyzed using DESeq2 with a two-sided Wald test. *p*-values were adjusted using the Benjamini–Hochberg method, and results were filtered for adjusted *p* < 0.05 and absolute log2 fold-change ≥0.5 using the ashr method. Non-FAP is white, normal FAP is ochre, and FAP adenomas are brown in the **b**, **d**, & **e** subdivision diagrams.

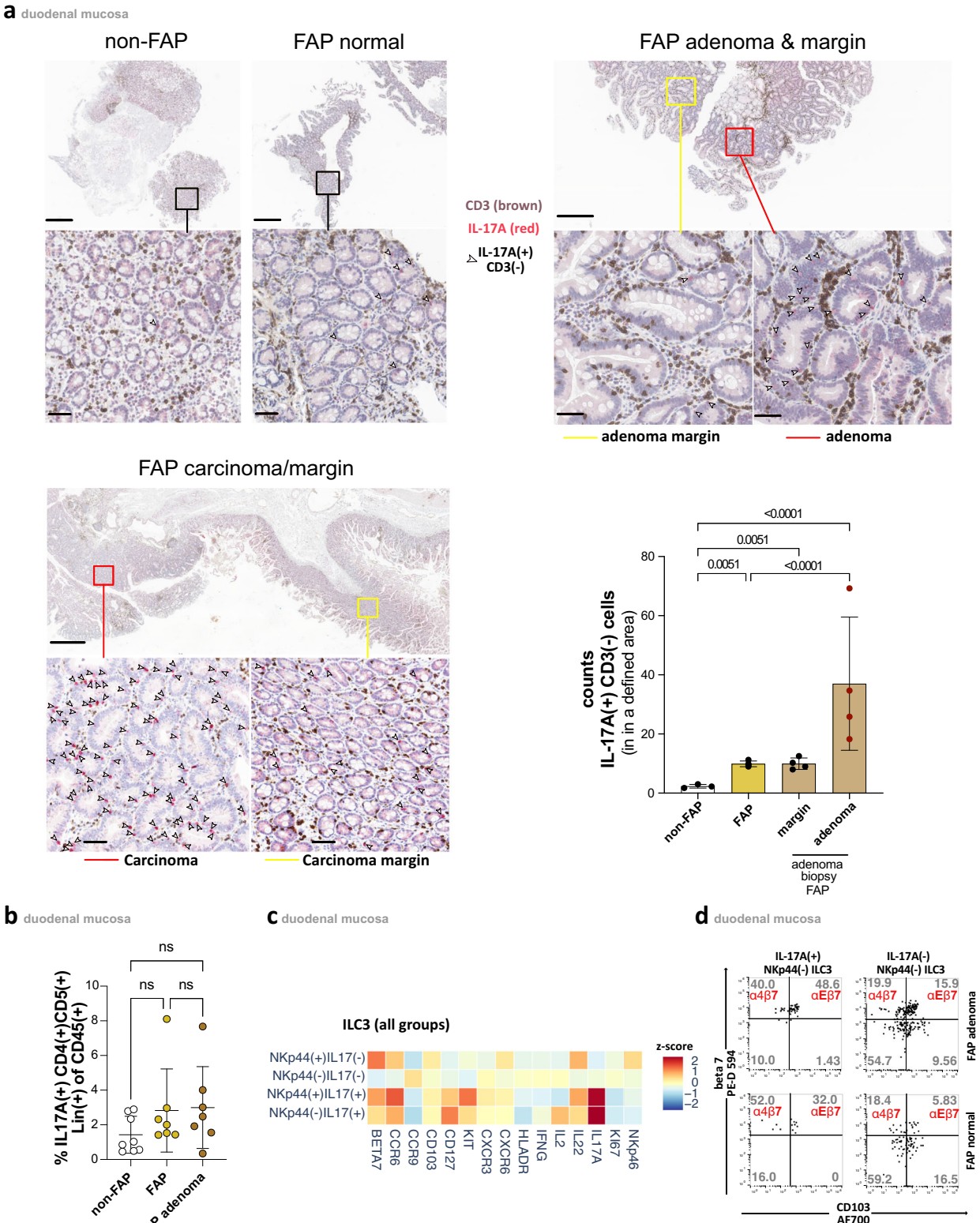

the primary producers of IL-17A. By employing this approach, we were able to spatially map CD3(−)CD19(−)NKp44(−)CD117⁺ ILC3 (Fig. 1f and Supplementary Fig. 3) and to confirm that these ILC3s are primarily located in the epithelial area, with a predominant presence in the subepithelial region as indicated by EPCAM co-staining.

In accordance with the data obtained from flow cytometry (Fig. 2c, d), immunohistochemistry (IHC) also confirmed an accumulation of IL-17A in FAP tissues, particularly within adenomas and FAP-

associated duodenal cancer tissue (Fig. 3a). The discovery of an elevated number of these cells in normal FAP tissue indicates that IL-17A(+) NKp44(−) ILC3s may begin to accumulate prior to oncogenic transformation, suggesting their potential involvement in the early stages of adenoma development.

Further phenotypic analyses demonstrated IL-17A(+)NKp44(−) ILC3 cells to display a CD56(-/+)NKp46(−)HLA-DR(−)CCR6(+) CCR9(low)CXCR6(low)CXCR3(−) phenotype (Fig. 3c and

**Fig. 3 | Epithelial localization and retention phenotype of IL-17A-producing ILC3s in FAP adenoma. a** Representative IHC stainings with IL-17A(red) and CD3(brown) of non-FAP(left panel), FAP normal (upper left panel), central (red square) adenomatous inclusive margin tissue (yellow square)(upper right panel), and FAP carcinoma (red square) inclusive margin tissue (yellow square)(lower left panel) with white arrows indicating CD3(−) and IL-17A(+)cells. Scale bars: 50 μm for magnified images, 500 μm for non-FAP/FAP-normal/FAP-adenoma & margin overviews, and 800 μm for FAP-carcinoma/margin overviews. Lower right panel: Counts of IL-17A( + )CD3(−) cells in IHC stainings including non-FAP ($n = 3$), FAP normal ($n = 4$), and FAP adenoma with defined margin and central adenoma region ($n = 4$), were analyzed. Mean values for each individual count were calculated from at least three regions of equal area of 440'000 μm$^2$ per biopsy sample. Mean ± SD. **b** Percentages of IL-17A( + )CD4( + )CD5(+)Lin(+) cells of CD45(+) cells in non-FAP ($n = 8$), normal FAP($n = 8$) and adenomatous FAP($n = 8$) tissue. Gating strategy of respective subsets in Supplementary Fig. 9e. Mean ± SD. **c** Heatmap showing $z$-score of flow cytometric evaluated expressions of indicated markers, gated on NKp44(+) and NKp44(−)ILC3s following PMA/Ionomycin stimulation, split into IL-17(−) and IL-17(+) subsets, from three subjects each of non-FAP, normal FAP, and FAP adenoma (blue-yellow-red scaling). Gating strategy of respective subsets in Supplementary Fig. 9c. **d** Representative dot plots of CD103 and β7 integrin from FAP normal and FAP adenoma patients on IL-17A(+) and IL-17A(−) NKp44(−)ILC3 following PMA/Ionomycin stimulation. CD103( + )β7(+) corresponds to αEβ7 expression, and CD103(−)β7(+) corresponds to α4β7 expression. Gating strategy of respective subsets in Supplementary Fig. 9d. Statistical significance analyzed by Two-way ANOVA (mixed model) with two-sided tests (**a**) and Kruskal–Wallis (KW) test (**b**) both corrected for multiple comparisons using FDR (Benjamini, Krieger, Yekutieli). $q$-values are indicated; ns = not significant. Non-FAP is white, normal FAP is ochre, and FAP adenomas are brown in the **a** & **b** subdivision diagrams.

Supplementary Fig. 4a, b). Importantly, their high expression of both αEβ7 and α4β7 integrins suggests strong retention within the lamina propria, particularly in subepithelial regions (Fig. 3d). In FAP adenomas, the IL-17A(+) NKp44(−) ILC3 cells showed increased expression of αEβ7 integrin, which may further enhance their adhesion to subepithelial areas. In contrast, the IL-17A(−) subset exhibited lower integrin expression and higher CCR9 levels (Supplementary Fig. 4b, c), indicating a closer association with deeper layers of the lamina propria.

In conclusion, our findings indicate that duodenal polyposis in FAP is associated with an increase in IL-17A(+)NKp44(−)ILC3s, predominantly localized in the epithelial area and marked by strong expression of epithelial retention markers. These observations suggest that these cells may shape a local duodenal microenvironment that predisposes the epithelium to malignant transformation in FAP.

### Increased IL-17A production of ILC3s in FAP is duodenum-specific

As colonic polyposis represents the main feature of FAP, we next tested whether alterations similar to those observed in duodenum can also be found in colon tissue. In line with previous reports[22], we found frequency of total ILCs as well as proportion of ILC3s to be significantly higher in the colon as compared to the duodenum (Supplementary Fig. 5a) with the vast majority of colon ILC3s being NKp44(+) (Supplementary Fig. 5b). Contrary to the duodenal compartment, no significant differences were observed in the frequencies of both total ILCs and group 3 ILCs between FAP and non-FAP colonic tissue (Fig. 4a, b and Supplementary Fig. 5). However, in colon adenomas, the proportion of NKp44(+)ILC3 in the total pool of colonic ILCs was reduced, while the proportion of ILC1 was increased (Fig. 4b, c).

Functional analysis of colon ILC3s did not reveal any significant differences regarding IL-17A production between FAP patients and controls (Fig. 4d), which again was in sharp contrast to our findings in the duodenum. Further analysis demonstrated frequencies of IL-17A-producing ILC3s to be significantly higher in the duodenum than in the colon in both normal and adenomatous tissues (Fig. 4e). However, these differences were found only in FAP patients and not in controls (Fig. 4f), which further supported the relevance of these cells for the pathogenesis of duodenal polyposis.

To summarize, these findings confirmed compartment-specific differences of intestinal ILC3s and demonstrated FAP-associated increase in IL-17A(+)NKp44(−)ILC3s to be duodenum-specific.

### Increased *IL1B/IL23A* and *DLL4* expression might induce IL-17A production of duodenal ILC3s in FAP

The local microenvironment critically regulates ILC biology. In particular, cytokines such as IL-1β and IL-23[24,25] and the Notch ligands Delta-like 1 and 4 (DLL1 and DLL4) have been shown to be important in the regulation of intestinal ILC3 differentiation, maturation, and function[26,27].

Therefore, we investigated whether altered mucosal cytokine and/or Notch ligand expression might be involved in duodenal accumulation and increased IL-17A production of ILC3s in FAP.

Comparing duodenal tissue from FAP patients and non-FAP controls, we observed significantly increased *IL1B* and *IL23A* expression in FAP, especially in duodenal adenomas. Importantly, both *IL23A* and *IL1B* mRNA levels were significantly correlated with frequency of duodenal IL-17A(+)NKp44(−)ILC3s (Fig. 5a, b).

Similar findings were made with respect to *DLL1* and *DLL4* expression, which was significantly increased in FAP duodenal tissue, with the highest expression in FAP adenomas, while only *DLL4* correlated significantly with IL-17A(+)NKp44(−)ILC3 frequency (Fig. 5c, d). Furthermore, MELC imaging demonstrated that NKp44(−)ILC3s were predominantly situated in close proximity to the duodenal epithelium in FAP patients, with multiple cells in close contact with CD31(+) endothelial cells that expressed DLL4 and DLL1 (Fig. 5e and Supplementary Fig. 6a; NKp44(−)ILC3 defined in Supplementary Fig. 6b). This intimate interaction suggests the existence of a niche where NKp44(−) ILC3s could be generated, potentially driven by signals from DLL4-expressing endothelial cells.

To further confirm a role for increased *IL1B/IL23A* and *DLL1/DLL4* expression in the regulation of ILC3 activity in FAP, we next performed in vitro bulk cell culture experiments. To this end NKp44(−) ILC3s were sorted on either OP9, DLL1-expressing (OP9-DL1) or DLL4-expressing (OP9-DL4) feeder cells. Analyzing the resulting supernatants after three days of culture in the presence of IL-1β and IL-23 demonstrated significant IL-17A concentrations only in the presence of OP9-DL4 feeder cells (Fig. 5f). To corroborate that the combination of DLL4 and cytokines is essential in regulating ILC3 activity in FAP, we used organoids to model conditions closer to the in vivo situation. Duodenal organoids derived from FAP patients, which exhibited high expression of DLL4 (Supplementary Fig. 6c), were co-cultured with ILC precursors (ILCP) in the presence of cytokines (IL1β, IL23, IL2, IL7). We observed that only this combination induced NKp44⁻ILC3s and sufficient IL-17A production (Supplementary Fig. 6d–f), confirming the data from our feeder cell experiments that neither DLL4 nor cytokines alone are sufficient, but their combination is necessary to induce ILC3 activity. Altogether, these data suggest increased levels of IL-1β and IL-23 together with increased DLL4 expression to be involved in the duodenal accumulation of IL-17A-producing ILC3s in FAP.

### Elevated *DUOX2* in FAP adenoma

Having shown an increased frequency of IL-17A-producing ILC3s in FAP duodenal tissue, we next aimed to explore the possible role of these cells in oncogenic transformation. As data on the molecular mechanisms involved in duodenal tumorigenic progression in FAP patients is limited, we first performed bulk transcriptome analyses to identify differentially expressed genes (DEG) in FAP duodenal tissue compared

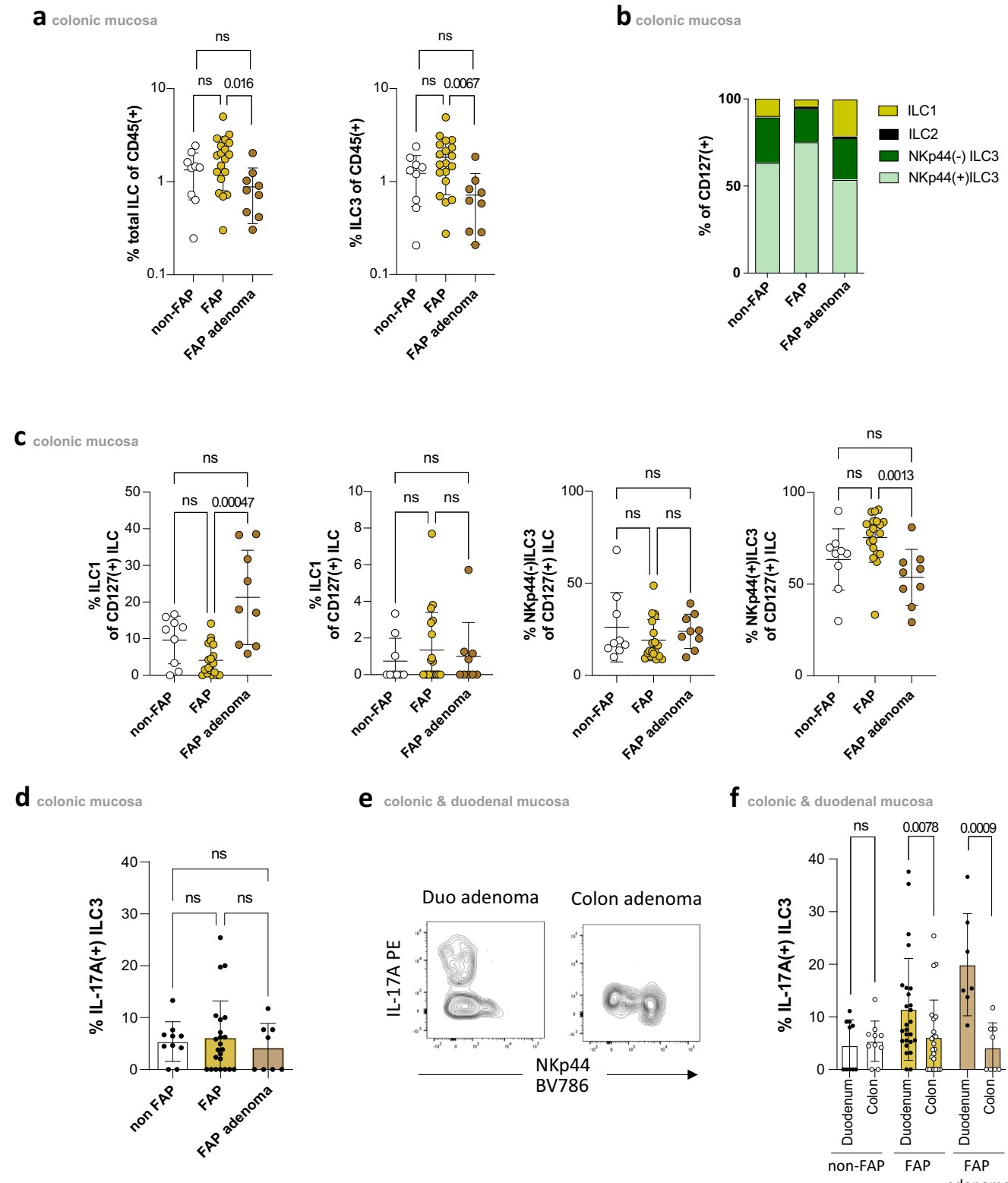

to non-FAP controls. Two-dimensional data representation using principal component analysis (PCA) demonstrated the separation of FAP adenoma from FAP normal tissue and control tissue (Fig. 6a). Differential expression analysis identified 161 upregulated and 84 downregulated genes when comparing FAP adenoma vs. non-FAP samples, and 112 upregulated genes and 48 downregulated genes when comparing FAP adenoma vs. FAP normal tissue (FC > |1.5|, $p$adj < 0.05; Fig. 6b, Supplementary Data 2). Among these, 103 genes

were consistently differentially expressed in both comparisons, comprising 76 upregulated and 27 downregulated genes, suggesting a consistent gene expression signature associated with malignant transformation in FAP patients (Fig. 6c and Supplementary Data 2). Several genes known to be involved in tumor formation and progression were significantly upregulated in FAP adenoma, including *CAPN8*, encoding for Calpain 8, a member of a family of intracellular calcium-activated neutral cysteine proteinases involved in cancer initiation,

**Fig. 4 | Increased frequency of IL-17A-producing ILC3s in FAP duodenal tissue.** **a** Frequency of total Lin(−)CD127(+)ILCs (left) and ILC3 (right) among CD45(+) cells in colonic adenomatous (*n* = 9) and normal mucosa (*n* = 19) of FAP patients and normal mucosa (*n* = 9) of non-FAP controls. Gating strategy of respective subsets in Fig. 1b. Mean ± SD. **b** ILC1(yellow), ILC2(blue), NKp44(−)ILC3(dark green), and NKp44(−)ILC3(light green) distribution in colonic adenomatous and normal mucosa of FAP patients and non-FAP controls. Gating strategy of respective subsets in Fig. 1b. **c** Percentages of ILC1, ILC2 and NKp44(+) and NKp44(−) colonic ILCs in colonic adenomatous (*n* = 9) and normal mucosa (*n* = 19) of FAP patients and normal mucosa (n = 9) of non-FAP controls. Gating strategy of respective subsets in Fig. 1b. Mean ± SD. **d** Percentages of IL-17A( +)ILC3 in colonic adenomatous (*n* = 8) and normal mucosa (*n* = 23) of FAP patients and normal mucosa (*n* = 10) of non-FAP

controls. Gating strategy of respective subsets in Supplementary Fig. 9a. Mean ± SD. **e** Representative image and **f** percentages of IL-17A (+)ILC3 in colonic and duodenal adenomatous (Duodenum: *n* = 7; Colon: *n* = 8) and normal mucosa (Duodenum: *n* = 26; Colon: *n* = 23) of FAP patients and normal mucosa (Duodenum: *n* = 10; Colon: *n* = 10) of non-FAP controls. Gating strategy of respective subsets in Supplementary Fig. 9a. Mean ± SD. Statistical significance analyzed by Kruskal–Wallis (KW) test (**a, c, d**) and Two-tailed Mann–Whitney test (**f**) corrected for multiple comparisons using FDR (Benjamini, Krieger, Yekutieli). q(KW)- and p(Mann–Whitney)-values are indicated; ns = not significant. Non-FAP is white, normal FAP is ochre, and FAP adenomas are brown in the **a, c, d,** & **f** subdivision diagrams.

progression, and metastasis[28], *TSPAN1*, encoding for Tetraspanin 1 which has been shown to promote growth of breast cancer cells via mediating PI3K/Akt pathway[29], *CEMIP*, which affects the WNT and EGFR signaling pathways and is involved in the progression of various tumors[30]. Among the genes with the highest magnitude of change were *CDH3* (P-Caherin), which has been found to be overexpressed in CRC[31], S100P, a calcium-binding protein P also overexpressed in CRC tissue[32] and *DUOX2*, a NADPH oxidase considered to also play a role in the development of various carcinomas[33]. The results were confirmed by qPCR, comparing transcripts in FAP adenomas with non-FAP controls (Fig. 6d). Of note, significant differences in *CEMIP*, *CDH3*, *DUOX2*, and *DUOXA2* levels were also found between normal FAP and non-FAP controls.

Enrichment and pathway analysis identified several upregulated pathways specifically in the "FAP adenoma vs. FAP normal" comparison, including the "hallmark estrogen pathway"[34] and "hallmark KRAS signaling"[35], both of which may play a role in tumorigenesis within the intestinal tract. Additionally, downregulation of the "WP energy metabolism" pathway, which plays a crucial role in tumorigenesis through the Warburg effect[36] was observed (Fig. 6e).

## IL-17A induces *DUOX2* and *DUOXA2* expression in duodenal organoids

Next, we tested whether IL-17A-producing ILC3 might be involved in establishing the transcriptional program observed in FAP adenomas, thereby promoting oncogenic transformation. To this end, we first generated duodenal organoids from normal duodenal mucosa of FAP patients and controls (Fig. 7a). Immunofluorescence analyses confirmed the presence of cells characteristic of the duodenum such as goblet and Paneth cells, as assessed by Muc-2 and Lyz (Fig. 7b), with no significant differences observed between organoids established from FAP and non-FAP specimens (Fig. 7c).

Stimulation of duodenal organoids with recombinant human IL-17A induced a significant upregulation of *DUOX2* and its maturation factor *DUOXA2*, whereas no such effects were observed for the other studied genes (Fig. 7d). Upregulation of *DUOX2* and *DUOXA2* induced by IL-17A was observed in organoids derived from both normal and adenomatous FAP duodenal mucosa, with no significant difference between the two (Supplementary Fig. 7a), and was also evident in colonic organoids, suggesting a general phenomenon (Supplementary Fig. 7b). Of note, none of the other ILC3-associated cytokines tested showed any impact on *DUOX2/DUOXA2* expression indicating an IL-17A-specific effect (Supplementary Fig. 7c).

Bulk-RNAseq analyses confirmed IL-17A-induced upregulation of *DUOX2* in both FAP and non-FAP organoids, whereas upregulation of *DUOXA2* was specifically observed in FAP organoids (Fig. 7e). Additionally, IL-17A upregulated a range of other tumor-associated genes and signaling pathways (Fig. 7f). Subsequent enrichment analysis revealed the activation of numerous pathways in IL-17A-stimulated FAP and non-FAP organoids (Fig. 7g). While both groups exhibited activation of cytokine-associated pathways consistent with IL-17 signaling, a notable difference was observed with FAP-derived

organoids uniquely showing significant enrichment in the "GO:BP Regulation of Hydrogen Peroxide Metabolic Process" pathway. This suggests that IL-17A specifically modulates oxidative stress pathways in FAP cells. Since DUOX2 and DUOXA2 are involved in hydrogen peroxide production, their upregulation further supports the notion that IL-17A influences oxidative stress mechanisms in the context of FAP.

## IL-17A-induced oxidative stress and Duox2-mediated DNA damage in FAP pathophysiology

To further substantiate these findings, we next tested the impact of duodenal ILC3 on *DUOX2/DUOXA2* expression in organoids. To this end, duodenal organoids were cultured with or without sort-purified and expanded duodenal NKp44(−)ILC3, NKp44(+)ILC3, or ILC1 in a medium containing recombinant human IL-1β and IL-23. *DUOX2 and DUOXA2* mRNA expression levels were only significantly increased in organoids cultured in the presence of NKp44(−)ILC3 (Fig. 8a). This effect was also seen when duodenal organoids were cultured in the presence of supernatants of NKp44(−)ILC3s expanded on OP9-DL4 (Fig. 8b), indicating a contact-independent mechanism. Accordingly, we found adding an IL-17A blocking antibody to prevent NKp44(−) ILC3-induced upregulation of *DUOX2* and *DUOXA2* whereas IL-17F blocking antibody could not prevent upregulation (Fig. 8c and Supplementary Fig. 8a). IL-17A induced increase in Duox2 expression in FAP adenoma and carcinoma tissues could also be corroborated at the protein level (Fig. 8d, e). Additionally, Duox2 protein expression was significantly upregulated after IL-17A stimulation in duodenal organoids at both one and three days post-treatment (Fig. 8f and Supplementary Fig. 8b). Given that Duox2 is known to generate reactive oxygen species (ROS) that can cause DNA damage, we hypothesized that its upregulation might lead to increased ROS production and subsequent DNA damage in FAP organoids.

To test this hypothesis, we conducted functional assays to quantify ROS production using DCFDA staining after 24 h and assessed DNA damage by measuring γH2AX levels via flow cytometry and immunofluorescence at 24 h and three days. Both ROS levels and γH2AX expression increased significantly in response to IL-17A stimulation (Fig. 8g–l and Supplementary Fig. 8b). Despite the increase in ROS production, MitoSOX staining indicated no rise in mitochondrial superoxide, suggesting that ROS is primarily extramitochondrial, driven by Duox2/DuoxA2 (Supplementary Fig. 8c). These findings emphasize the role of IL-17A in regulating oxidative stress in FAP organoids, primarily through Duox2/DuoxA2, which may result in DNA damage and, thereby, contributing to FAP pathophysiology.

To determine whether IL-17A exerts additional or more general effects on duodenal organoids beyond pathways related to oxidative stress and DNA damage, we assessed the expression of genes characteristic of duodenal epithelial cells, as well as markers associated with proliferation and angiogenesis. IL-17A stimulation increased MUC-2 expression but did not alter the expression of EpCAM, LYZ (lysozyme), or LGR5, genes typical of duodenal epithelial and stem cells, suggesting that IL-17A does not broadly affect the differentiation

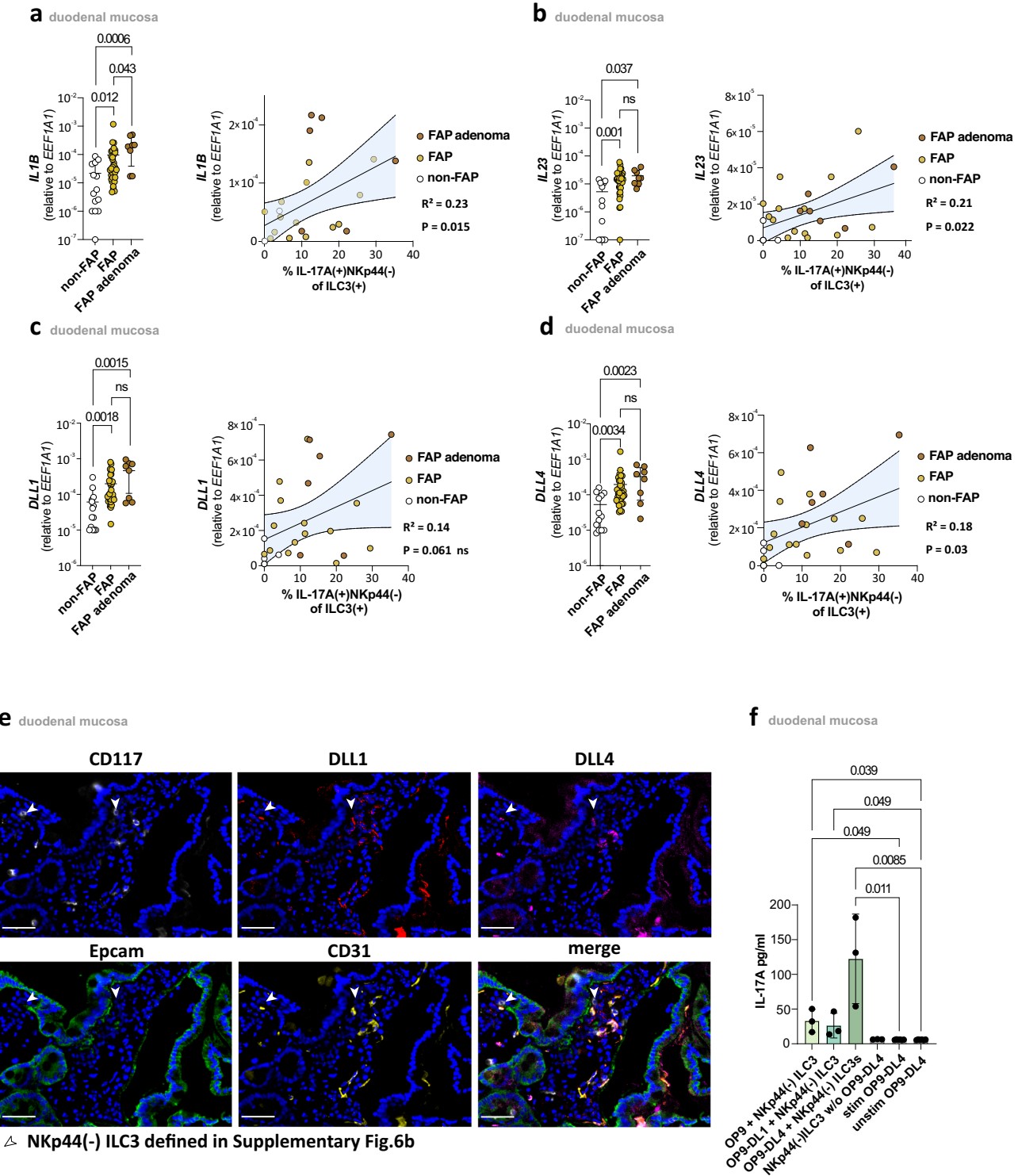

**e** duodenal mucosa

△ NKp44(-) ILC3 defined in Supplementary Fig.6b

status of duodenal cells (Supplementary Fig. 8d). Similarly, there were no significant changes in the expression of HIF1α, Ki-67, or β-Catenin, which are markers associated with hypoxia response and cell proliferation.

However, IL-17A significantly upregulated the transcript level of VEGFA, a key factor in angiogenesis (Supplementary Fig. 8d). Thus, in addition to inducing oxidative stress and DNA damage, IL-17A may promote angiogenic pathways, potentially contributing to oncogenic transformation by enhancing vascularization.

Overall, these findings emphasize that IL-17A does not have a general effect on duodenal organoids but specifically influences pathways related to oxidative stress, DNA damage, and angiogenesis.

## Discussion

In addition to colonic polyposis and CRC, duodenal adenomas are the most prevalent intestinal manifestation of FAP, significantly increasing the risk of duodenal cancer compared to the general population[9]. Nevertheless, the varying duodenal phenotype and clinical

**Fig. 5 | Increased duodenal expression of *IL1B*, *IL23A*, *DLL1*, and *DLL4* in FAP.**
**a** Left panel: *IL1B* mRNA expression in duodenal adenomatous ($n = 8$) and normal mucosa ($n = 33$) of FAP patients and normal mucosa ($n = 14$) of non-FAP controls. Mean ± SD. Right panel: Pearson correlation between duodenal *IL1B* mRNA expression and the proportion of IL-17A-producing NKp44(−) cells within total ILC3s ($n = 25$). **b** Left panel: *IL23A* mRNA expression in duodenal adenomatous ($n = 8$) and normal mucosa ($n = 33$) of FAP patients and normal mucosa ($n = 14$) of non-FAP controls. Mean ± SD. Right panel: Pearson correlation between duodenal *IL23A* mRNA expression and the proportion of IL-17A-producing NKp44(−) cells within total ILC3s ($n = 25$). **c** Left panel: *DLL1* mRNA expression in duodenal adenomatous ($n = 8$) and normal mucosa ($n = 33$) of FAP patients and normal mucosa ($n = 14$) of non-FAP controls. Mean ± SD. Right panel: Pearson correlation between duodenal *DLL1* mRNA expression and the proportion of IL-17A-producing NKp44(−) cells within total ILC3s ($n = 25$). **d** Left panel: *DLL4* mRNA expression in duodenal adenomatous ($n = 8$) and normal mucosa ($n = 33$) of FAP patients and normal mucosa ($n = 14$) of non-FAP controls. Mean ± SD. Right panel: Pearson correlation

between duodenal *DLL4* mRNA expression and the proportion of IL-17A-producing NKp44(−) cells within total ILC3s ($n = 25$). mRNA expression levels are expressed in relation to mRNA expression levels of *EEF1A1*. **e** Representative MELC images of FAP normal tissue showing NKp44(−)ILC3 cells, indicated by white arrows, with CD117(white), DLL1(red), DLL4(violet), EpCAM(green), CD31(yellow) and nucleus stainings(blue). ILC3 cells are defined in Supplementary Fig. 6b as CD45( + ) CD117( + )CD3(−)CD14(−)NKp44(−)CD16(−)EpCAM(−) lymphoid cells. White scale bar represents 20 μm. Based on three independent biological replicates. **f** IL-17A concentration in the supernatant of tonsil NKp44(−)ILC3 and controls (each 3 donors) after 3 days of culturing on OP9, OP9-DL1, and OP9-DL4 cells or unstimulated and stimulated OP9-DL4 alone ($n = 6$ replicates), respectively. Mean ± SD. Colors as indicated. Statistical significance analyzed by the Kruskal–Wallis (KW) test (**a**, **b**, **c**, **d**, **f**) corrected for multiple comparisons using FDR (Benjamini, Krieger, Yekutieli). *q*-values are indicated. Pearson correlation with corresponding $R^2$ and *p*-values is shown. Non-FAP is white, normal FAP is ochre and FAP adenomas are brown in the **a**, **b**, **c**, & **d** subdivision diagrams.

progression among carriers of the same genetic variant suggest that factors beyond the genotype, such as the local immune system, contribute to these differences[8,12].

Besides their function in providing immunity against pathogens and preserving tissue balance at mucosal locations[17], recent findings indicate that ILCs also have a significant role in tumor development and progression. However, the precise contribution of ILCs to gastrointestinal tumors, particularly duodenal tumorigenesis, remains incompletely understood, and the available data on their involvement are limited.

Here, we present evidence suggesting that ILC3 may play a role in the development of duodenal polyposis in FAP. First, we show that FAP is associated with a significantly increased frequency of NKp44(−)ILC3 in the duodenum. Importantly, ILC3 numbers were highest in adenoma tissue but were already increased in macroscopically normal mucosa, suggesting that FAP-associated duodenal ILC3 infiltration precedes adenoma development. Second, we found not only ILC3 frequencies but also duodenal ILC3 production of the tumor-promoting cytokine IL-17A to be increased in FAP patients. Third, in vitro studies demonstrated that IL-17A and duodenal ILC3s induce *DUOX2/DUOXA2* expression, with the former specifically upregulating Duox2 protein, thereby driving mitochondrial-independent ROS production and DNA damage, a key mechanism in cancer progression[33,37]. Together, these findings suggest that NKp44(−)ILC3 plays a role in duodenal oncogenic transformation in FAP.

The development and function of ILCs are critically affected by the local microenvironment, with ILC3 being dependent on IL-23 and IL-1β[15]. We found FAP to be associated with significantly elevated *IL23A* and *IL1B* mRNA levels not only in duodenal adenoma but also in macroscopically normal duodenal tissue. This was an interesting observation as levels of IL-23 and its receptor chains have been shown to be elevated in numerous human cancers, including CRC, and to correlate with disease progression[38]. Accordingly, mice lacking the IL-23p19 chain were found to rarely exhibit intestinal tumorigenesis[39], whereas minicircle-based systemic overexpression of IL-23 was sufficient to induce de novo adenoma formation in the duodenum without other exogenous triggers. Mechanistically, it was postulated that IL-23 promotes inflammatory responses and increases angiogenesis but reduces CD8 T-cell infiltration[38]. Our data suggest that modulation of ILC3 responses may represent an additional mechanism by which IL-23 may promote tumor formation as we not only observed duodenal *IL23A* levels to positively correlate with frequencies of mucosal IL-17A-producing ILC3 but also demonstrate IL-23 to trigger expression of the cancer-promoting cytokine IL-17A in duodenal ILC3. IL-23-induced upregulation of the Notch ligands DLL1 and DLL4 may play a role in this context, as we observed increased *DLL1* and *DLL4* mRNA levels in adenomatous tissue. In our in vitro experiments, IL-17A-producing NKp44(−)ILC3s were exclusively induced in the presence of DLL4-

expressing OP9 cells or organoids. Additionally, NKp44(−)ILC3s were found in situ near DLL4( + )CD31(+) endothelial cells, indicating significant interaction.

However, we observed that DLL4 alone is not sufficient to induce IL-17-producing ILC3s. Additional cytokines, such as IL-23 and IL-1β, were necessary, suggesting that these cytokines likely have a more direct effect on ILC3 induction. Altogether, these findings point to the significance of both DLL4 and IL-23/IL-1β in the increased presence of duodenal NKp44(−) ILC3s in FAP.

The involvement of IL-17A(+)ILC3 in malignant transformation resembles data obtained in mice. Chan et al. demonstrated mice lacking adaptive immune cells and ILCs (RAG2[-/-]IL-2Rγc[-/-]) to be resistant to tumor formation triggered by IL-23, whereas tumor development did not differ between RAG1[-/-] and wild-type mice[40]. In line with our findings, IL-17A was proposed to contribute to IL-23-induced adenoma development of duodenal tumors in these models, as IL-17A expression was found to be increased in mice developing tumors. In addition, Rag1[-/-]IL-17A[-/-] mice also displayed protection against tumor formation[40], further supporting a role for IL-17A-producing ILCs.

We demonstrated in vitro that IL-17A leads to an increased expression of *DUOX2*, which corresponds with the elevated *DUOX2* expression observed in FAP adenomas. This suggests a potential role of DUOX2 in ILC3-mediated oncogenic transformation, aligning with other studies that demonstrate IL-17A-induced *DUOX2* upregulation and its role in ROS-mediated pathogenesis[41–44].

Duox2 belongs to the family of NADPH oxidases (NOX). It forms a complex with its maturation factor, DuoxA2, to catalyze the synthesis of hydrogen peroxide ($H_2O_2$). Duox2 acts as the first barrier of the intestinal epithelium and is involved in the innate immune response of the intestinal mucosa[45], but it also plays a role in the development of various carcinomas. Increased *DUOX2* expression has been found in liver cancer[46], pancreatic cancer[47–49], and prostate cancer[50]. In addition, *DUOX2* was found to be highly expressed in CRC and to promote CRC cell invasion and metastasis by affecting the ubiquitination status of ribosomal protein uL3[33]. Moreover, Duox2 has also been shown to affect the response to gastrointestinal cancer treatment[51,52]. In this study, Duox2 expression was found to be significantly higher in flat colon polyps than in adjacent normal epithelial tissues[53], suggesting that overexpression of Duox2 may promote polyp occurrence. These findings, along with our results, suggest that Duox2 may contribute to the early stages of intestinal tumorigenesis, especially by increasing ROS production and subsequent DNA damage. FAP may represent a unique context in this process, as its genetic background could enhance susceptibility to ROS-related damage. Of note, in patients with FAP, the loss of APC removes a key regulatory mechanism that normally mitigates ROS levels, promoting cellular repair processes or apoptosis. The absence of APC may contribute to the persistence of

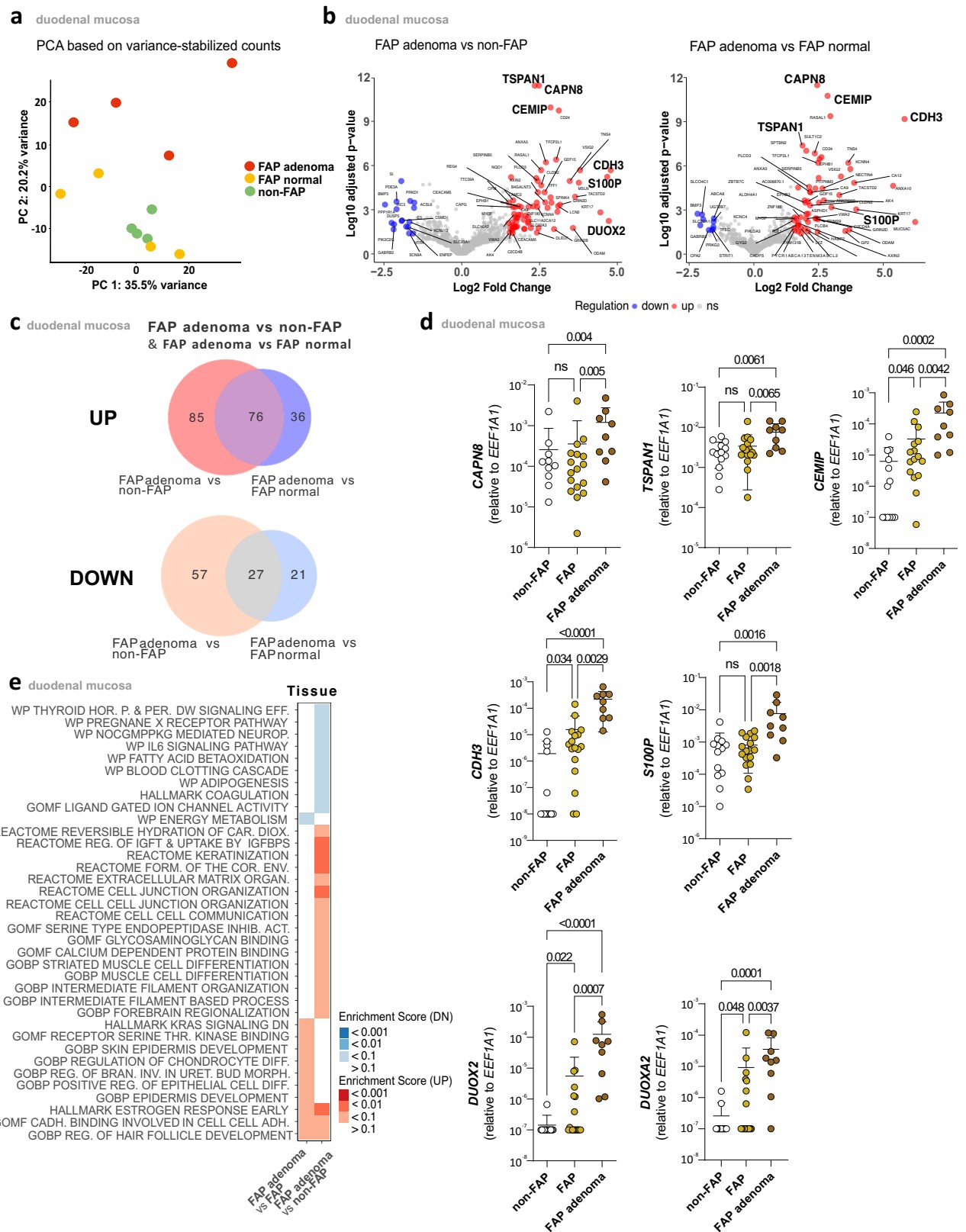

ROS, potentially leading to further DNA damage and, thereby, promoting oncogenic transformation[54].

Our data indicate an important role for IL-17A-producing ILC3s in FAP adenomas, however, we did not perform a detailed analysis of the entire immune infiltrate. Nevertheless, our findings suggest that ILC3s, rather than Th17 cells, are the primary source of IL-17A, as most IL-

17A(+) cells in FAP adenomas were CD3(−). Additionally, the frequency of Th17 cells did not significantly differ between groups, supporting the conclusion that Th17 cells are not the dominant contributors to IL-17A production in this context.

Other ILC subsets, such as CD4(+) ILC1, CD5(+) mature ILC1, immature ILCs, and LT-ILC1[55], may also play important roles in FAP-

**Fig. 6 | Transcriptome analysis of duodenal tissue in FAP vs. non-FAP controls.**
**a** Principal component analysis of bulk RNA-seq data from duodenal normal(-yellow) and adenomatous mucosa of FAP patients(red) and normal mucosa of non-FAP controls (green). Each point represents the gene expression profile of a single sample, with colors indicating different patient groups. **b** Volcano plot displaying log2 fold-changes (FC) and FDR-adjusted *p*-values comparing adenomatous mucosa of FAP patients and normal mucosa of non-FAP controls (left panel). Differentially expressed transcripts (log2 FC > |1.5|, *p*(adj) <0.05) are highlighted in red or blue, depending on up- or downregulation in adenomatous mucosa, respectively. Additionally, the right panel shows a comparison between FAP adenoma and FAP normal mucosa. **c** Venn diagram illustrating the overlap of up- and downregulated differentially expressed genes (DEGs) between two comparisons:

FAP adenoma vs. FAP normal(blue) and FAP adenoma vs. non-FAP normal(red). **d** qPCR results comparing gene expression in duodenal normal (*n* = 17) and adenomatous (*n* = 9) mucosa of FAP patients and normal mucosa (*n* = 13) of non-FAP controls. mRNA expression levels are relative to *EEF1A1* expression. Mean ± SD. **e** Bulk RNA-seq analysis showing a heatmap of the enrichment scores (up&down) for significant pathways, based on comparisons of FAP adenoma vs. FAP normal and FAP adenoma vs. non-FAP normal(blue-red scaling). Error bars represent SD. Statistical significance analyzed by the Kruskal–Wallis (KW) test corrected for multiple comparisons using FDR (Benjamini, Krieger, Yekutieli). *q*-values are indicated; ns = not significant. Non-FAP is white, normal FAP is ochre, and FAP adenomas are brown in the **d** subdivision diagrams.

related tumorigenesis and should be explored in future studies to further clarify the immune landscape of FAP adenomas.

In conclusion, our data indicate a role for IL-17A-secreting NKp44(−)ILC3 in duodenal tumorigenic progression in FAP. These findings highlight the role of local immune responses in the early stages of duodenal cancer development associated with FAP, offering potential avenues for innovative therapeutic strategies.

## Methods

### Ethical statement
This research complies with all relevant ethical regulations and adhered to the Declarations of Helsinki and Istanbul and received approval from the University of Bonn ethics committee (#079/13, #040/16, #275/13 and #493/20). Informed consent was secured from all participants.

### Human tissue samples
During routine endoscopy, macroscopically normal duodenal tissues were collected from FAP patients with *n* = 101, 49/101 female, age 39.4 (16–81) and non-FAP controls with *n* = 42, 25/42 female, age 47.4 (15–75) as well as duodenal adenomas from FAP patients with *n* = 32, 18/32 female, age 40.5 (17–73) and a FAP patient with duodenal carcinoma with *n* = 1, female, age 65–70.

Control samples: normal colon tissue from FAP patients with *n* = 24, 13/24 female, age 27.8 (16–69) and non-FAP controls with *n* = 10, 7/10 female, age 40.4 (23–62), colonic adenomas from FAP patients with *n* = 13, 7/13 female, age 23 (20–39), peripheral blood (*n* = 4) and post-tonsillectomy human tonsils (*n* = 6). Detailed information is given in Supplementary Data 1.

### Lymphocyte isolation
Intestinal biopsies[14] were incubated with pre-digestion medium for 45 min at 37 °C, further digested for 60 min at 37 °C in digestion medium, and filtered through a 70 μm cell strainer (Supplementary Table 2). Tonsils were cut, squeezed through a metal sieve, and centrifuged using Pancoll gradient centrifugation. PBMCs were isolated using density gradient centrifugation. Isolated cells were pre-frozen in a freezing medium (Supplementary Table 2) at −80 °C and stored at −150 °C until further use. Thawing was performed using a thawing medium (Supplementary Table 2).

### Stimulation of lymphocytes
Lymphocytes (200,000) were stimulated with PMA (50 ng/ml) and Ionomycin (1 μg/ml) (P/I) in complete RPMI (Supplementary Fig. 1) for 4 h, with BFA added after 1 h for flow cytometry. Unstimulated cells were used as controls.

### Bulk cell cultures
Sorted circulating or tonsil NKp44(−)ILC3 with identical cell counts used across all conditions in a comparative approach were added to 96 well-plate with or without OP9, OP9-DLL1, and OP9-DLL4 feeder cells,

kindly provided by Dr. Marcus Uhrberg, Prof. Dr. Juan Carlos Zuniga-Pflucker and Prof. Dr. Andreas Diefenbach and incubated with differentiation medium (Supplementary Table 2). Every 2–3 days, the medium was changed, and IL-2, IL-23, IL-7, and IL-1β (10 ng/ml) were added (Supplementary Table 1). After short- or long-term culture, the resulting NKp44(−)ILC3s were analyzed or sorted. Supernatants were analyzed using Legendplex.

### Organoid culture
Organoids were generated from intestinal samples using the IntestiCult protocol (#06010, STEMCELL). Samples were washed with PBS, minced and centrifuged, then incubated in Gentle Cell Dissociation Reagent (GCDR, STEMCELL) for 30 min. After centrifugation, DMEM + 1% BSA was added, and crypts were pipetted out. 1000 crypts/matrigel dome were plated in a 24-well plate, with Organoid Growth medium and gentamicin (50 μg/ml) added and changed every 2–3 days (Supplementary Table 1). After 5–7 days, organoids were splitted using GCDR, analyzed by immunofluorescence, flow cytometry, bulkRNA-seq, or qRT-PCR.

For immunofluorescence, bulkRNAseq or qRT-PCR organoids were harvested using Corning® Cell Recovery Solution for 30 min on ice and pelleted (Supplementary Table 1). For flow cytometry analysis, organoids were dissociated using TrypLE (Thermo Fisher Scientific) for 20 min at 37 °C and stained as previously described[22].

### Cytokine stimulation, co-culture, and blocking experiments
Organoids were stimulated with 10 ng/ml cytokines or supernatants obtained from NKp44(−)ILC3 cultures after 5 days. Isotype control or IL-17A/F blocking antibodies (Supplementary Data 3) were added to supernatants as indicated. After 20 h, the organoids were harvested for RNA isolation, flow cytometry, or immunofluorescence.

In co-culture experiments following established protocols[56], duodenal NKp44(−)ILC3 or circulating NKp44(−)ILCPs were sorted, placed with identical cell counts used across all conditions in a comparative approach within the organoid dome, and stimulated every two days with 10 ng/ml IL-1β, IL-23, IL-2, and IL-7 (Supplementary Table 1). After 5-7 days, co-cultures were lysed for qRT-PCR analysis or stained for flow cytometry. Supernatants were preserved for Legendplex analysis.

### Organoid staining for flow cytometry
Organoids were cultured in 24-well plates for 5 days and then stimulated with IL-17A for 20 h. After 20 h, the cells were dissociated, stained with Zombie Aqua, and fixated with paraformaldehyde (PFA) for 10 min (Supplementary Table 1). After washing, pre-cooled 70% ethanol (AppliChem) was added drop by drop while vortexing and incubated 4 °C overnight. Cells were washed, permeabilized in 0.25% Triton X-100 (in 0.5% BSA PBS) for 5 min, and washed again (Supplementary Table 1). Blocking was performed in PBS containing 1% BSA for 10 min. Cells were then split for controls and staining. Organoids were stained with γH2AX2 primary antibody and Duox-2

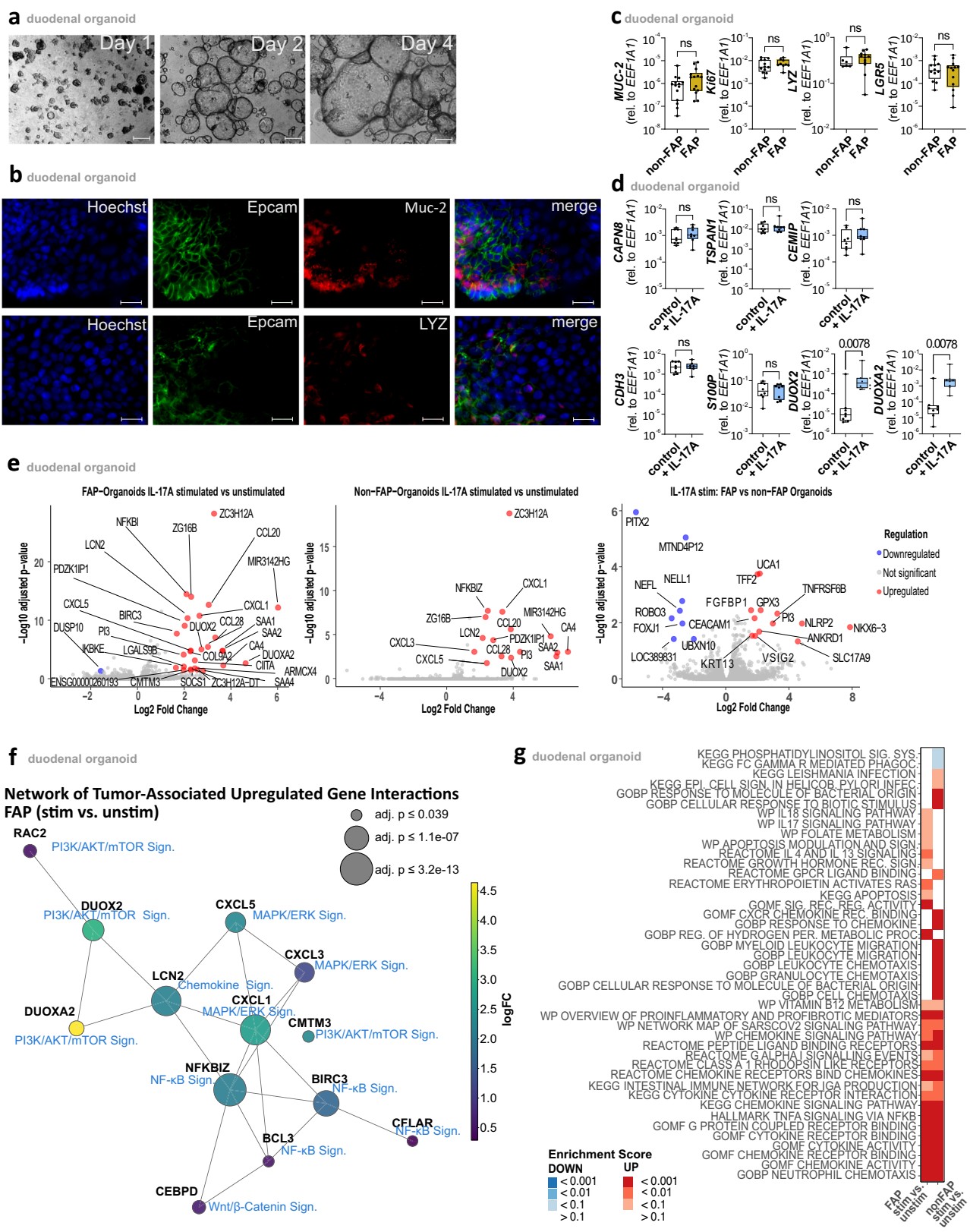

specific antibody (1:200) (Supplementary Data 3). After washing, anti-mouse-Alexa-Fluor 488 and rb555 secondary antibody was applied (Supplementary Data 3). Cells were washed and stained with anti-KI67 BV421 and EpcamAF647 for 30 min, except for the control samples (Supplementary Data 3). Cells were washed with PBS for FACS analysis.

## ROS production

The levels of intracellular ROS were analyzed using 20,70-dichlorodihydrofluorescein diacetate (DCFDA; Abcam)(Supplementary Table 1). Organoids were cultured in 24-well plates for 5 days and then stimulated with IL-17A for 20 h. After 20 h, the cells were dissociated and incubated for 20 min at 37 °C with 100 nM DCFDA. After 10 min,

**Fig. 7 | IL-17A increased *DUOX2/DUOXA2* expression in duodenal organoids.**
**a** Duodenal organoids on day 1, 2, and 4. Distance is indicated in a 200 μm bar. Based on five independent biological replicates. **b** Hoechst(blue), Lysozyme (LYZ) (red), Mucin-2 (Muc-2)(red) and Epcam(green) immunofluorescence staining in duodenal organoids. Distance is indicated in a 20 μm bar. Based on five independent biological replicates. **c** *MUC-2* (non-FAP, $n = 14$; FAP, $n = 14$), *KI67* (non-FAP, $n = 12$; FAP, $n = 9$), *LYZ* (non-FAP, $n = 6$; FAP, $n = 12$) and *LGR5* (non-FAP, $n = 12$; FAP, $n = 13$) mRNA expression in duodenal organoids of non-FAP controls(white bar) and FAP patients(ochre bar). Box plot showing the median (center line), interquartile range (box bounds: 25th–75th percentile), and range (whiskers: min–max). **d** qPCR results comparing gene expression of duodenal organoids ($n = 6$) with(white bar) or without 20 h IL-17A stimulation(blue bar). mRNA expression levels are relative to *EEF1A1* expression. Box plot showing the median (center line), interquartile range (box bounds: 25th–75th percentile), and range (whiskers: min–max). **e** Volcano plots displaying log2-fold-changes (FC) and FDR-adjusted *p*-values comparing FAP organoid IL-17A stimulated vs. unstimulated (left panel), non-FAP organoid IL-17A stimulated vs. unstimulated (middle panel), and IL-17A stimulated condition: FAP vs. non-FAP organoids (right panel). Differentially expressed transcripts (log2 FC > |1.5| , p(adj) < 0.05) are highlighted in red or blue, depending on up- or downregulation in the comparisons, respectively. **f** Gene-pathway network visualization of differentially expressed genes in IL-17A stimulated vs. unstimulated duodenal FAP organoids. Nodes represent individual genes, with sizes proportional to adjusted *p*-values and colors indicating log fold-changes (blue-yellow scaling). Edges denote known protein-protein interactions based on STRING database data (see Supplementary Table 2). **g** Bulk RNA-seq analysis showing a heatmap of the enrichment scores (up & down) for significant pathways, based on comparisons of FAP organoid IL-17A stimulated vs. unstimulated and non-FAP organoid IL-17A stimulated vs. unstimulated (up: white-red scaling; down: white-blue scaling). Statistical significance was analyzed by two-tailed Mann–Whitney test (**c**) and Two-tailed Wilcoxon matched-pairs signed rank test (**d**). *p*-values are indicated; ns = not significant. DEGs (**e**–**g**) were analyzed with a two-sided moderated *t*-test. *p*-values were adjusted for multiple comparisons using the Benjamini–Hochberg method.

antibodies and Zombie Aqua were added for 10 min at 37 °C (Supplementary Table 1). Subsequently, the cells were washed and analyzed by flow cytometry.

## BrdU assay
Organoids were cultured for 5 days and then stimulated with IL-17A for 20 h. After 20 h, the cells were dissociated and treated with 10 μM BrdU for 17 h (Supplementary Table 1). After 17 h, the Cytofix/CytopermTM Kit was used to permeabilize (Supplementary Table 1). Subsequently, single cells were incubated with DNAseI for 30 min at 37 °C. Anti-BrdU PE and Zombie Aqua were added for 20 min at 37 °C. After washing, the cells were analyzed by flow cytometry.

## MitoSox staining
Duodenal organoids were cultured for 5 days and then stimulated with IL-17A (10 ng/ml), IL-17A (10 ng/ml) + rotenone (positive control), or left untreated as a control, in organoid medium for 20 h. Subsequently, the cells were washed again and stimulated with MitoSOX (Supplementary Table 1) for mitochondrial reactive oxygen species (mROS) detection for 30 min at 37 °C. After stimulation, the organoids were dissociated and washed. The cells were then incubated with Zombie Aqua, followed by staining with anti-EpCAM-BV421 (Supplementary Data 3). Finally, the cells were analyzed using FACSCanto II and FlowJo software V10.7.1.

## Flow cytometry analysis and cell sorting
Flow cytometric analyses of cells were performed using an LSR-Fortessa Cytometer (BD, Germany) and a spectral Sony ID7000 7-laser cytometer[22]. In brief, all antibodies were titrated, the panels were tested using FMO controls, and constant conditions were ensured by plate staining to guarantee an optimal staining result. The antibodies used in these studies are listed in Supplementary Data 3. Zombie Aqua/ Zombie NIR excluded dead cells, and intracellular analysis used the Invitrogen Foxp3 Transcription Factor Kit and Cytofix/Cytoperm™ Kit (Supplementary Table 1). Data was analyzed via FlowJo Software V10.7.1. Alternatively, the Cytolytics software (cytolytics.de) was used for heatmap generation. Here, raw data were transformed using the Asinh transformation, followed by batch correction with Cytocorrect (cytolytics.de). The analysis involved gating on the relevant subsets, and a Z-score for the expression of the markers depicted in the figures was calculated and visualized in a heatmap. Cell sorting was performed using a FACSAria™ Fusion cell sorter (BD Bioscience).

## Cytokine detection assay
Cytokine secretion was assessed by cytokine bead array (LEGEN-Dplex™ Human Th Cytokine Panel Kit, Biolegend®), including data acquisition on a BD FACSCanto II and analysis using the software provided by the manufacturer.

## Immunofluorescence staining
Organoids growing on a coverslip in a 48-well plate or cut tissue sections were fixed with 4% PFA (Merck), washed, and incubated with a blocking buffer (Supplementary Table 2) for 30 min at RT. Organoids were incubated with primary antibodies (Supplementary Data 3) overnight at 4 °C. After washing, the secondary antibody (Supplementary Data 3) was added for 45 min at RT. The nucleus was stained with 1:1000 dilution Hoechst (Thermo Fisher Scientific) for 5 min at RT. After washing steps, slides were mounted with Prolong™ Gold antifade reagent (Invitrogen™) and analyzed with a fluorescence microscope DM IL (Leica Microsystems GmbH) or a Axio Imager.Z2 (Zeiss). Images were analyzed using ImageJ software (NIH, version 1.2) or Qupath 0.5.

Duox2 protein quantification in biopsy specimens was performed on digital images captured using a Zeiss Axiovert 220 M microscope equipped with a mercury lamp. Duox2 expression was analyzed by measuring Duox2-positive area per cell using ImageJ software (NIH, version 1.2), with cell boundaries defined by Hoechst staining. For each biopsy specimen, the mean Duox2-positive area was calculated by averaging measurements from all cells across a minimum of five randomly selected microscopic fields. For the quantification of Duox2 and γH2AX2 signals in organoids, images were acquired using the Zeiss software ZEN 3.6 blue edition, and TIFF files were generated for the channels corresponding to Hoechst, Duox2, and γH2AX2. For the detection of γH2AX2, cell counts for each organoid were automatically determined using the Cell Counter tool in Qupath 0.5 software, based on nuclear staining with Hoechst. Subsequently, all γH2AX2-positive cells within the selected area were manually counted, and the percentage of γH2AX2-positive cells was calculated relative to the total cell count. For the detection of Duox2, separate staining was performed on organoids. Using Hoechst nuclear staining and autofluorescence, the cell area for each detected cell was defined. The mean fluorescence intensity of Duox2 was measured for each cell within the defined area. Mean values for Duox2 were calculated from all intensities for all cells in an organoid or defined area of a biopsy and are presented in the "Source data & Results section".

## IHC microscopy
2–3 μm thick sections of FFPE material were cut on TOMO slides. The slides were then placed in the BenchMark Ultra (Fa. Roche), and the deparaffinization, heat pretreatment and the complete further reaction were carried out in the device. The antigen demasking pretreatment was performed by boiling the slides in CC1 buffer at a pH 8. For the double staining, the automatic protocol from BenchMark Ultra (Fa. Roche) was used, antibodies against CD3 (NCL-L-CD3-565, Fa. Leica, Clone LN10, dilution 1:50), IL17A (AB79056, Fa. Abcam, dilution 1:200), Duox2 (NOVUS, NB110-61576, dilution 1:200) and following detection

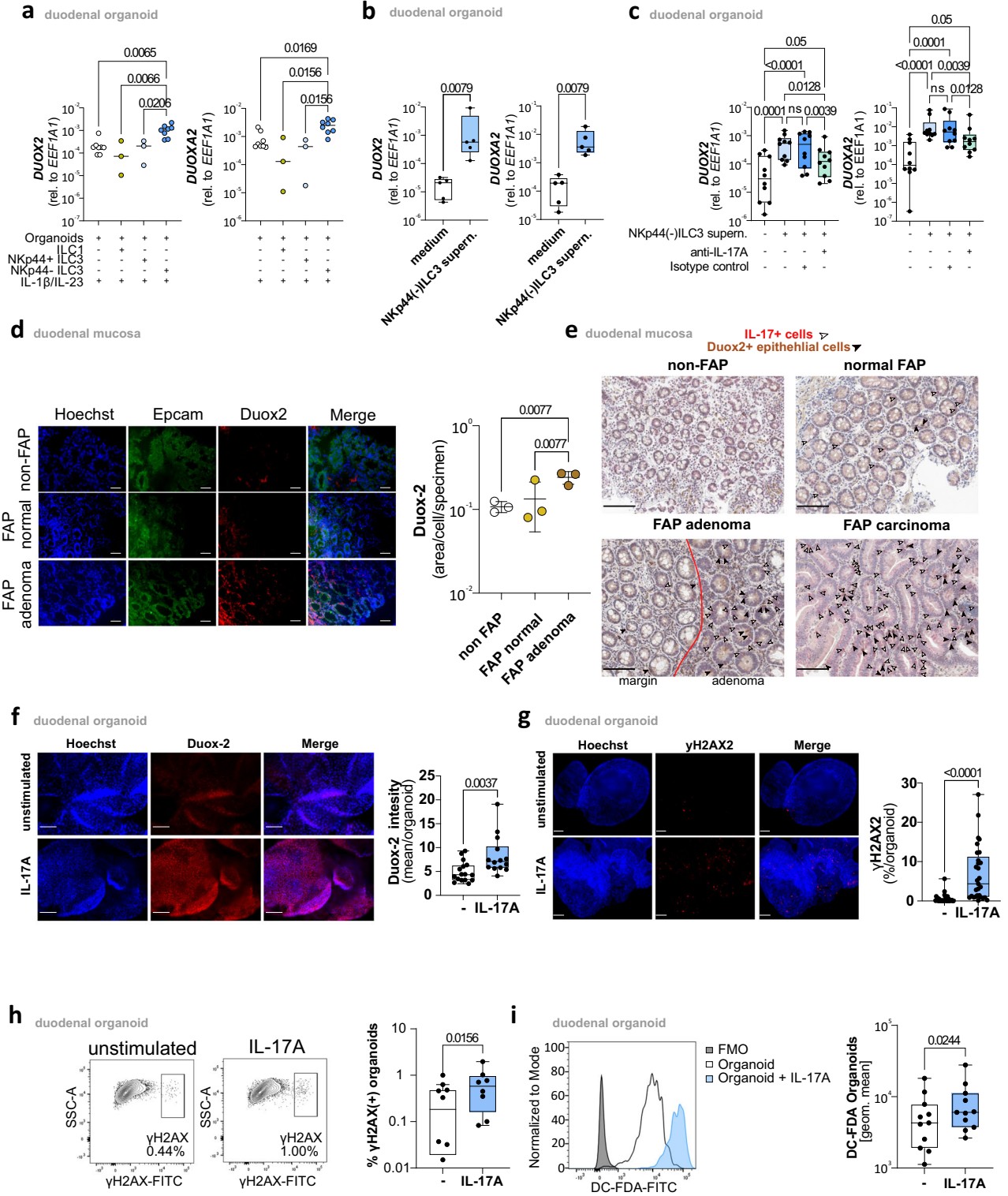

kits: UltraView / OptiView DAB detection kit (Roche, 760-700) and UltraView Universal Alkaline Phosphatase Red Detection Kit (Roche, 760-501). The slides were then counterstained with Mayer hemalaun, dehydrated using an ascending alcohol series, and covered with an anhydrous mounting medium. High-resolution histological images of the combinations CD3/IL-17A and Duox2/IL-17A were captured using a Leica Biosystems Aperio. Central adenoma regions were distinguished

from adenoma margins in biopsies, using criteria such as the absence of goblet cells and increased nuclear density in the epithelial compartment. The analysis was conducted using Qupath 0.5 software.

**Multi-epitope ligand cartography (MELC)**

**Tissue preparation for MELC.** Human duodenum samples were prepared as described here[57]. In short, fresh frozen samples were put in

**Fig. 8 | IL-17A-Induced oxidative stress and Duox2-mediated DNA damage in FAP pathophysiology. a** *DUOX2* and *DUOXA2* mRNA expression in duodenal organoid cultures incubated alone (*n* = 8)(white) or in the presence of NKp44(+) ILC3 (*n* = 3)(light blue), ILC1 (*n* = 3)(yellow), or NKp44(−)ILC3 (*n* = 8)(blue) and within the matrigel for 4 days. Every other day 10 ng/ml IL-23A and IL-1β were added to the supernatant in each condition. **b** *DUOX2* and *DUOXA2* mRNA expression in duodenal organoid cultures (*n* = 10) incubated in the presence(blue) or absence(white) of supernatants of NKp44(−)ILC3s on OP9-DL4 feeder cells. Box plot showing the median (center line), interquartile range (box bounds: 25th–75th percentile), and range (whiskers: min–max). **c** *DUOX2* and *DUOXA2* mRNA levels were measured in duodenal organoid cultures (*n* = 10) incubated(light blue) with or without(white) NKp44(−)ILC3 supernatants in the presence or absence of an IL-17A-specific blocking antibody(green) or isotype control(blue). Box plot showing the median (center line), interquartile range (box bounds: 25th–75th percentile), and range (whiskers: min–max). **d** Representative IF images with Duox2(red), Hoechst(nucleus)(blue), and EpCAM(green), and with merged images (left panel, scale bar: 50 μm). Right panel: mean Duox2 staining intensity (normalized to nuclear area with Hoechst) across non-FAP(white bar), FAP(ochre bar), and FAP adenoma tissues (brown bar) (*n* = 3 per group, technical replicates are listed in the Source Data table). **e** Representative IHC images showing Duox2(brown) and IL-17A(red) in non-FAP, normal FAP, FAP adenoma with margin, and FAP carcinoma tissue. White arrows indicate IL-17A(+)cells, and black arrows indicate Duox2(+) epithelial cells. Scale bar represents 100 μm. Based on five independent biological replicates. Red line marks the boundary between margin and central adenoma. **f** Left panel: representative IF staining of Duox2(red) and Hoechst(nucleus)(blue) in duodenal organoids, unstimulated and IL-17A-stimulated for three days. Scale bar represents 50 μm. Right panel: Duox2 mean intensity relative to Hoechst staining, with unstimulated (*n* = 17 organoids)(white) and IL-17A-stimulated samples (*n* = 14 organoids)(blue). Box plot showing the median (center line), interquartile range (box bounds: 25th–75th percentile), and range (whiskers: min–max). **g** Left panel: representative IF staining of γH2AX(red) and Hoechst(nucleus)(blue) in duodenal organoids, unstimulated and IL-17A-stimulated for three days. Scale bar represents 50 μm. Right panel: frequency of γH2AX(+)cells, with unstimulated samples (*n* = 31 organoids)(white) and IL-17A-stimulated organoids (*n* = 29 organoids)(blue). Box plot showing the median (center line), interquartile range (box bounds: 25th–75th percentile), and range (whiskers: min–max). **h** Left panel: representative flow cytometric staining of γH2AX in duodenal organoids, unstimulated and IL-17A-stimulated for one day. Right panel: frequency of γH2AX(+) cells, with unstimulated (*n* = 8)(white) and IL-17A-stimulated organoid cultures (*n* = 8)(blue). Box plot showing the median (center line), interquartile range (box bounds: 25th–75th percentile), and range (whiskers: min–max). **i** Left panel: representative flow cytometric staining of DC-FDA for ROS detection in duodenal organoids, unstimulated(white) and IL-17A-stimulated(blue) for one day, with FMO(Fluorescence minus One) control(gray). Right panel: geometric mean intensity of DC-FDA staining, with unstimulated (*n* = 11)(white) and IL-17A-stimulated organoid cultures (*n* = 11)(blue). Box plot showing the median (center line), interquartile range (box bounds: 25th–75th percentile), and range (whiskers: min–max). Statistical significance analyzed by Two-tailed Mann–Whitney test (**b, f, g**), Two-tailed Wilcoxon matched-pairs signed rank test (**h, i**), Kruskal–Wallis (KW) test (**a**), and Two-way ANOVA (mixed model) with two-sided tests (**d**) and Two-tailed Friedman test (**c**) corrected for multiple comparisons using FDR (Benjamini, Krieger and Yekutieli). q(KW, Friedman and ANOVA-mixed model)- and p(Mann–Whitney, Wilcoxon)-values are indicated; ns = not significant.

Tissue-Tek cryomolds (Sakura) filled with O.C.T. medium, and frozen using 2-Methylbutane (Sigma Aldrich, St. Louis, USA) and liquid nitrogen. Samples were cut in 5 μm sections using a NX80 cryotome (ThermoFisher, Waltham, Massachusetts, USA) and placed on 3-aminopropyltriethoxysilane (APES; Sigma Aldrich)-coated cover slides (24 × 60 mm; Menzel-Gläser, Braunschweig, Germany). Sample fixation was done for 10 min at room temperature with freshly prepared electron microscopy grade 2% paraformaldehyde (methanol- and RNAse-free; Electron Microscopy Sciences, Hatfield, Philadelphia, USA) solution. After three rounds of washing with PBS, permeabilization was performed with 0.2% Triton X-100 in PBS for 10 min at room temperature. The fluid chamber holding 100 μl of PBS was created using "press-to-seal" silicone sheets (Life technologies, Carlsbad, California, USA; 1.0 mm thickness).

**MELC image acquisition.** For MELC data acquisition, a BioDecipher® Device 1.0 (BioDecipher GmbH, Magdeburg, Germany) was used that was equipped with the following components: (1) ORCA®-Flash4.0 V3 Digital CMOS camera (C13440-20CU; Hamamatsu Photonics GmbH, Japan) acquiring images of 2048 × 2048 pixels (pixel size 6.5 μm, no binning), (2) Leica DMi8 microscope (Leica Microsystems GmbH, Wetzlar, Germany) with a (3) Leica HC PL APO 20x/0.80 PH 2 objective lens, and a (4) Cavro® XLP 6000 Pump (Tecan GmbH, Crailsheim, Germany). The MELC panel was designed using the device control software (BioDecipher GmbH, Magdeburg, Germany). Antibodies were titrated to find the best working concentration in human duodenum tissue, with detailed information provided in Supplementary Data 3. Prior to each MELC experiment, PBS with 5 % MACS BSA (Miltenyi Biotec, Bergisch Gladbach, Germany) was prepared, and Köhler illumination was performed.

**Image preprocessing.** MELC is based on sequential incubation-acquisition-photobleaching cycles automatically performed by a pipetting robot and has been described for murine and human tissues before[58–60]. Image preprocessing was done using the TIC OBSERVER software (BioDecipher GmbH, Magdeburg, Germany),

and normalization with ImageJ 1.2[61] as described in Pascual-Reguant et al.[60].

## PCR analysis
For qRT-PCR, frozen intestinal tissue samples and organoids were thawed and messenger RNA (mRNA) was extracted using the GeneJet RNA purification Kit (Thermo Scientific). cDNA was transcribed using the QuantiTect reverse transcription Kit (Qiagen) in accordance with the manufacturer's protocol. qRT-PCR was performed on a Light-Cycler 96 (Roche) with the Blue S'Green qPCR Kit (Biozym®) using the primer sets depicted in Supplementary Table 3. Ct values were analyzed with the LightCycler 96 software version 1.1.0.1320, the. The relative gene expression was calculated by the $2^{-\Delta Cq}$-method using the expression of the housekeeping gene *EEF1A1* as a reference for normalization.

## Bulk RNA-sequencing and transcriptome analysis
All information regarding bulk RNA-seq for the tissue, organoid, and sorted ILC3 datasets is provided in Supplementary Data 2.

## Statistical analysis and reproducibility
After testing for normal distribution, differences between the two groups were evaluated using the unpaired Wilcoxon–Mann–Whitney test or the paired Student's *t*-test, as appropriate. For comparisons among three or more groups, the Kruskal–Wallis test was employed or a mixed-effects model was used to account for variability within sub-samples when necessary, both with adjustments for multiple comparisons using the Benjamini, Krieger, and Yekutieli false discovery rate (FDR) control method[62]. Correlations between two variables were tested by Pearson-test. Statistical analyses were performed using Prism version 9.4.0 (GraphPad Software).

Transcriptome analysis was performed with R (v. 3.6.2 or 4.0.3): the specific packages used for the analysis, their version, and relevant parameters used are provided in Supplementary Data 2. Plots and heatmaps were generated with ggplot (v. 3.3.2) and complexheatmap (v. 2.2.0).

All software, tools, algorithms, and packages are listed in Supplementary Table 4.

## Reporting summary
Further information on research design is available in the Nature Portfolio Reporting Summary linked to this article.

## Data availability
The bulk RNA-seq raw data supporting this study are deposited in the European Genome-phenome Archive (EGA) under study accession EGAS00001007347 and dataset accession EGAD00001015488. Access is restricted in accordance with European data protection regulations (GDPR) to protect participant privacy. Researchers requesting access must contact the Data Access Committee (DAC) via the EGA website. Requests are reviewed based on compliance with participant consent. Approved users must sign a Data Access Agreement outlining confidentiality and restrictions on redistribution. The DAC typically responds within a few weeks; upon approval, data access is granted for an unlimited period. Further details are provided on the EGA dataset page. All relevant code required for bulk RNA-seq tissue analyses is available at the following GitHub repository: https://github.com/Ulaslab/FAP_paper_Nattermann. Source data are provided in this paper. All the other data are available within the article and its Supplementary Information. To comply with patient consent restrictions and mitigate the risk of participant identification, gender was excluded from patient characteristics in Supplementary Data 1, and age was categorized into 5-year intervals. Notably, our analysis confirmed that these factors showed no association with important study parameters. These data were presented during the review process. Source data are provided in this paper.

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

## Acknowledgements

J.N., T.U., A.C.A., F.S., M.H., and K.T. are members of the excellence cluster ImmunoSensation (EXC 2151) funded by the German Research Foundation (DFG) under grant agreement #390873048; DFG SPP1937 [to J.N.]; HA 5354-8/2 [to A.E.H.]; SFB 1444 P14 [to A.E.H.]; KR 4521/1-1 [to B.K.]; SFB 1454—432325352 [to A.C.A.] and the Hector foundation [M88 to J.N.]. Further funding was provided through the German Center for Infection Research (DZIF). We thank PRECISE at DZNE & University of Bonn for sample pro-cessing and RNA sequencing. We would like to thank the Flow Cytometry and Bioinformatics Core Facility of the Medical Faculty at the University of Bonn for providing support and instrumentation funded by the DFG—project number 387333827, 216372401, 01EO2107 (BMBF). We would like to thank the Core Unit for Bioin-formatics Data Analysis of University Hospital Bonn for providing computing resources and support in setting up the initial pipeline. We would like to thank Tillmann Wehner and Laura Jimena Ruiz, participants in the Bundesfreiwilligendienst, for their valuable support. We are grateful to the patients and donors volunteering to participate in this study making this research possible. Figure 1a includes images from Servier Medical Art, licensed under CC BY 4.0.

## Author contributions

Conception & design: K.M.K., R.H., B.K., J.N. Methodology: K.M.K., B.K., G.H., N.M., J.C., S.T., M.H., S. Ka, J.N. Investigation: K.M.K., R.H., B.K., J.N. Data analysis: K.M.K., R.H., G.H., J.C., B.K., J.N. Data production & ana-lysis-bulkRNA-seq: E.D.C., M.D.B., T.U., A.C.A., M.B., A.B., F.S., K.T., L.M.S., D.W., K.H., S.N., O.B., C.G. Data processing & analysis MELC: S. Kr, R.U., A.E.H. Sample collection and processing: K.M.K., J.R., M.T.V., G.H., S.A., N.M., S.I.W., J.C., P.A., L.A.G., D.J.K., T.M., F.G., H.D.N., P.L., N.S., T.V., M.T., S.S., O.H., J.O., A.M., J.E., F.N.G., F.I.S., P.Z., C.P.S., R.H., B.K., J.N. Writing—original draft: K.M.K., B.K., J.N. Writing—review & editing: all authors.

## Funding

## Competing interests

R.H. and J.N. received endoscopic equipment on loan from Fujifilm, Germany. Robert Hüneburg has received consulting fees for medical advice from CPP-FAP, One Two Therapeutics, and Janssen Pharmaceu-ticals. C.P.S. has received speaker honoraria from Falk Pharma, Eisai, Astellas, Chiesi, MSD, and support for seminars from Gilead, Bristol-Myers Squibb, Abbvie, MSD, Norgine, Tillots Pharma, Eisai, Janssen, Falk Foundation, and consulting fees for medical advice from Fa. Schwabe, Astra Zeneca, Eisai and Astellas. The remaining authors declare no competing interests.

## Additional information

[1]Department of Internal Medicine I, University Hospital Bonn, Bonn, Germany. [2]National Center for Hereditary Tumor Syndromes, University Hospital Bonn, Bonn, Germany. [3]Department of Surgery, University Hospital Bonn, Bonn, Germany. [4]Department of Pathology, University Hospital Bonn, Bonn, Germany. [5]Institute of Experimental Haematology and Transfusion Medicine, University Hospital Bonn, Bonn, Germany. [6]Institute of Experimental Oncology, University Hospital Bonn, Bonn, Germany. [7]Department of Otorhinolaryngology, University Hospital Bonn, Bonn, Germany. [8]Department for Otorhinolaryngology, Head and Neck Surgery, University Medical Center Mainz, Mainz, Germany. [9]Gastroenterology, Bonn, Germany. [10]Institute for Medical Biometry, Informatics and Epidemiology, Medical Faculty, University of Bonn, Venusberg-Campus 1, 53127 Bonn, Germany. [11]Institute for Genomic Statistics and Bioinformatics, Medical Faculty, University of Bonn, Venusberg-Campus 1, 53127 Bonn, Germany. [12]Systems Medicine, Deutsches Zentrum für Neurodegenerative Erkrankungen (DZNE), Bonn, Germany. [13]Genomics and Immunoregulation, Life & Medical Sciences (LIMES) Institute, University of Bonn, Bonn, Germany. [14]Immunogenomics & Neurodegeneration, Deutsches Zentrum für Neurodegenerative Erkrankungen (DZNE), Bonn, Germany. [15]PRECISE Platform for Single Cell Genomics and Epigenomics, DZNE, University of Bonn, and West German Genome Center, Bonn, Germany. [16]Systems Biology of Inflammation, German Rheumatism Research Center (DRFZ), Leibniz Association, Berlin, Germany. [17]Department of Rheumatology and Clinical Immunology, Charité-Universitätsmedizin Berlin Corporate Member of Freie Universität Berlin and Humboldt-Universität zu Berlin, 10117 Berlin, Germany. [18]Immune Dynamics, Deutsches Rheuma-Forschungszentrum (DRFZ), a Leibniz Institute, 10117 Berlin, Germany. [19]Institute of Innate Immunity, Medical Faculty, University of Bonn, Bonn, Germany. [20]Division of Infectious Diseases and Tropical Medicine, University Hospital, LMU Munich, 81377 Munich, Germany. [21]German Center for Infection Research (DZIF), Bonn, Germany. [22]These authors contributed equally: Robert Hüneburg, Benjamin Krämer, Jacob Nattermann. ✉e-mail: benjamin.kraemer@ukbonn.de; jacob.nattermann@ukbonn.de

