## [Transparent Peer Review file · Nature Communications]

IL-17A-producing NKp44(-) group 3 innate lymphoid cells accumulate in Familial Adenomatous Polyposis duodenal tissue

Corresponding Author: Professor Jacob Nattermann

Version 0:

Reviewer comments:

Reviewer #1

(Remarks to the Author)

This study focused on investigating the role of intestinal innate lymphoid cells (ILCs) in familial adenomatous polyposis (FAP), an inherited gastrointestinal tumor syndrome associated with an increased risk of duodenal cancer. The researchers analyzed duodenal and colon samples from FAP and non-FAP patients using flow cytometry and RNA analysis. The findings revealed that FAP patients had a higher frequency of NKp44- ILC3s, in both normal and adenomatous duodenal tissue from these patients. These ILC3s were found to produce increased levels of IL-17A, and increased levels of IL-1 β , IL-23 and DLL4 in the adenomas may be involved in duodenal accumulation of IL-17A-producing ILC3s. The higher presence of this subset and IL-17A production correlated with elevated expression of certain genes related to inflammation and cancer development. Overall, the study suggests that IL-17A-producing NKp44- ILC3s may play a role in the formation of duodenal adenomas in FAP. While IL-17 production is commonly associated with NKp44- ILC3 cells, particularly during gut inflammatory circumstances, IL-17A expression under normal conditions has been rarely observed in adult tissues. Additionally, IL-17A is not produced by tonsil ILC3s, and the transcript is very low in adult ileum ILC3s. Thus, the increase of these population in the duodenum of FAP patients shows a uniqueness to this niche, which favors the development of these subsets. However, without a more comprehensive mechanistic understanding, it is challenging to definitively conclude that IL-1 β , IL-23, and DLL4 are directly responsible for the observed accumulation of IL-17A-producing ILC3s in FAP. Further information on the patients used in the study as well as experiments that employ more advanced techniques are warranted to establish a causal link and unravel the precise mechanisms driving this phenomenon.

Major points:

1 – The authors propose that increased levels of IL-1 β and IL-23, along with increased DLL4 expression, are involved in the duodenal accumulation of IL-17A-producing ILC3s in FAP. While the correlation between these factors is intriguing, the paper falls short in providing a robust mechanistic understanding of this relationship. The study relies on correlational data and in vitro experiments, which are limited in elucidating the causal mechanisms at play. To strengthen their argument, it would have been beneficial for the authors to conduct functional studies that directly evaluate the impact of the niche on the accumulation of IL-17A-producing ILC3s in relevant in vivo models or ex vivo systems. For instance, the authors could co-culture with the organoids from FAP patients with CD34+ HSP or ILCPs, using the previously described protocols.

2 - Further information on the demographic data of the patients included in this study and their treatment status is warranted. The wide range of ages among the patients raises questions about potential age-related variations in the proportion of NKp44- ILC3s, especially given the previous observations that this population is reduced in adults. There is clearly a range of IL-17A production by these cells among the FAP patients, as well.

It would be valuable to present this information in a separate table, alongside the corresponding percentages of NKp44- ILC3s of CD45+ cells and % of IL-17A NKp44- ILC3s. Such data would enable a more thorough analysis of any potential correlations between patient demographics and the prevalence of this specific ILC3 subset and their function. It is also important to clarify whether the patients included in this study were receiving any specific treatments at the time of sample collection. Treatment regimens have the potential to impact the immune cell composition and function, thus potentially confounding the interpretation of the results.

3 – The authors primarily emphasize the production of IL-17A by NKp44- ILC3s, yet it is essential to consider the potential involvement of Th17 cells, which are also known to produce IL-17A. It would be important for the authors to investigate whether Th17 cells are increased and whether they exhibit higher IL-17A production.

4 – Although the authors show that blocking IL17A in the culture supernatants reduced the upregulation of DUOX2 and DUOXA2, it would be relevant to further understand the function of these cells. For instance, similar to the experiment in Figure 5, the authors could sort NCR+ and NCR- ILC3s separately from patient samples and check for DEGs between these subsets. This analysis could also shed light into potential crosstalk between NCR- ILC3s with other cell types as well as to a more in-depth phenotyping of NKp44- ILC3s.

5 – In addition, the authors show the surface markers expressed by the ILC3 cluster but fail to show how the expression of each marker as well as the cytokine production differs between each of the ILC3 subsets. For instance, are the IL-17A+ NCR- ILC3s preferentially CD56 negative, as previously reported?

5 - The authors report that stimulation of duodenal organoids with IL-17A specifically induced upregulation of DUOX2 and DUOXA2, while no significant effects observed for other studied genes. Although this observation provides insight into the potential role of IL-17A in the regulation of DUOX2 and DUOXA2, the limited analysis of gene expression leaves important questions unanswered. To gain a more comprehensive understanding of the transcriptomic changes induced by IL-17A, it would be beneficial for the authors to perform bulk RNA-seq analysis of the organoids after exposure to IL-17 or the supernatant from NCR- ILC3s culture. If the latter is chosen, it would be crucial to stimulate the organoids with the supernatant of other ILC cell culture, such as NKp44+ ILC3s, not just media as in Figure 7B. This could reveal additional genes or pathways influenced by IL-17A signaling in the context of duodenal adenoma formation in FAP.

Minor points

1 – In Supplementary 2B, correct CD200R for CD200R1 instead.

2 - In Supplementary 2B, CD117 is higher on normal tissue and decreases in FAP. The authors should clarify how was ILC3s gated on the samples when CD117 was lower. Do they have additional markers to confirm they were gating on ILC3s?

Reviewer #2

(Remarks to the Author)

Kaiser et al report on the role of ILCs in duodenal adenomas. The authors report collection of normal duodenal tissue from FAP patients (n=82, 40/82 female, age 17-84) and non-FAP patients (n=26, 12/26 female, age 15-72), as well as duodenal polyps from patients (n=16, 10/16 female age 16-72). Furthermore, they collect control samples: normal colon tissue from FAP patients (n=26, 14/26 female, age 18-67) and non-FAP controls (n=11, 6/11 female, age 24-58), colonic adenomas from FAP patients (n=14, 7/14 female, age 19-43).

To study ILCs in the lesions, they enzymatically dissociate adenoma biopsies, isolate mononuclear cells, and analyze them by FACS using an extensive dump gate, and differentiating the three major types of ILCs (1, 2 & 3) base on expression of CD127 (IL-7 receptor) and CD117 (cKIT; ILC3) and CRTH2 (Prostaglandin DP2 receptor; ILC2). They distinguish t two subsets of ILC3s as NCR (Natural Killer Cell Receptor) negative or positive based on their expression of NKp44. They find increased densities of ILC1s and ILC3s but not ILC2s in the FAP adenomas, relative to healthy tissue from FAP or non-FAP patients. Among ILC3 the increase was restricted to the NKp44 negative population. The show that a higher frequency NKp44 negative population produces more IL17a in the adenomas but find no difference for NKp44 positive population of the ILCs. Also, expression of IL17 was restricted to the duodenum polyp ILCs and was not obvious among the colon polyp ILCs. Interestingly, the vast majority of healthy colon ILC3s were NKp44 positive, and overall, the colon had more ILCs than the small bowel. Using bulk RNAseq they observed increased expression of IL1b, IL23, and the Notch ligands Delta-like 1 and 4 (DLL1 and DLL4) in duodenal polyps. They also demonstrate a correlation with increased expression of IL17A and IL23 or DL1 or DL4. To better understand the relation between these the culture NKp44 negative ILCs in the presence of IL23 and IL1b and show that addition of dl4 increases expression of IL17a by the cells. Stimulation of the organoids with IL17a induced expression of DUOX2 (dual oxidase 2). DUOX2 expression by polyp organoids was upregulated in response to IL17a or co-culture with ILC3s. The authors conclude that the ILC3s have a role in the growth of duodenal adenomas.

This is the first study of ILCs in human duodenal adenomas and adds substantially to our knowledge of the role of ILCs in small bowel cancer. The findings are not completely in line with an earlier study by Sonnenberg and colleagues (PMID: 34407392) which reported down regulation of ILC3s in tumors of CRC patients. The authors do not quote this article. Also, the findings contrast with similar studies done in mice and are therefore somewhat surprising. The authors report that the NCR- rather than the NCR+ ILC3s expand in polyps and have proinflammatory properties. This contrasts with the proposed Treg-inducing role of NCR- ILC3s and the proinflammatory potential of the NCR+Tbet+ ILC3s in mice. Additional work is needed to establish whether this difference between human and mouse ILC3s is real. The work is done very carefully and with rigor. However, the conclusion that ILC3s contribute to polyp growth is premature and not strongly supported by the provided data. The story would have been complete had the authors shown that the NCR- ILC3s or supernatant of these cells enhance polyp growth and/or invasive properties. This could be done using an ex vivo assay or better also using xenogeneic tumor transplant in immune-deficient mice. As it stands the manuscript is of high interest, but additional data is needed to validate the conclusions.

Reviewer #3

(Remarks to the Author)

In this manuscript Kaiser KM et al., claim that their data indicate a role for IL17A secreting NKp44- ILC3 in duodenal adenoma formation in FAP, but I do not think that the results support this conclusion. The only 2 differences that the Authors show between normal FAP and adenoma FAP is that ILC3 are increased and that IL17A+ILC3 are increased but only if considered as percentage of CD45+ cells. I think that what the Authors show is that IL17A+ILC3 are associated with FAP in general.

Major:

Since CD45+ cells do not change, is the ILC1 and ILC3 increase due to proliferation? Or they were recruited in the intestine? How is the expression of gut homing receptors in these 2 populations? Are the ILC1 functional?

Figure 2B and E: The Authors state that IL17A production was most prominent in adenoma tissue. However, there is no difference between FAP and FAP adenoma. Thus, the IL17A increase could be a shared phenomenon of FAP. The Authors claim that there is no difference in phenotype, however in the Supp Figure 2 CD56 seems upregulated in FAP adenoma. How is the quantification? How is the expression of NKp46? Also, ILC3 were characterized as cells expressing CCR6 but negative for CXCR3. Are the ILC3 identified in this way different in frequency between FAP and FAP adenoma? CCR6+NKp46+ ILC3 are a source of IL-22 and IL-17A. The Authors should evaluate this population. Also, how is the expression of ROR γ t, T-bet and c-Maf? Are these cells more or less prone to become proinflammatory ILCs producing IFN- γ ?

OP9 experiment: I think that this experiment is not designed in the correct way (and the methods do not match with what is written in the results). OP9 stromal cells are used as feeders either to differentiate ILCs from CD34+ precursors, or to generate ILC clones. In this work, according to what is written in the results, NKp44- ILC3 (bulk?) were incubated for 3 days with cytokines and different OP9. First, if the Authors wanted to show the impact of Notch ligands, they could have added recombinant proteins expressing only the extracellular portion of Dll1 or 4, instead than the OP9. Second, in the first column w/o ILC3 which are the cells producing IL17A, the OP9, OP9DL1 or OP9DL4? IL17A secretion in the presence of only IL1b and IL23 (or with medium alone) is a control that need to be shown. In any case, in my opinion, the Authors should do single cell cloning with ILC3 obtained from non-FAP, FAP and FAP adenoma using the different OP9 to compare the percentages of clone formation and the identity of the clones. To confirm the identity of the clones ICS including transcription factors and type1, 2 and 3 cytokines should be used.

In patients, which cells are expressing DLL1 or DLL4?

RNAseq: The Authors show only the difference between FAP adenoma and control. What is the difference between FAP and FAP adenomas?

The Authors only focused on IL-17A, but in the last experiment with the IL-17A blocking antibody it is clear the this is not the only player. What about IL-17F for instance? The fact that ILC3 supernatant induces DUOX2 or its active metabolite DUOXA2, which as per the Authors "are thought to promote cancer cell appearance and progression" does not prove that they actually do it. Moreover, only one gene (and an associated one) is not indicative of any change. How are these cells behaving in tumour invasion assays? Do they migrate/proliferate more? It would be important to have specimens from duodenal cancer patients to verify whether ILC3, IL17A and DUOX2 or DUOXA2 (and others) are increased. In vitro, it would be informative to show whether NKp44- (and not NKp44+) ILC3, in an IL17A dependent manner are able to convert human colonic adenoma cells (such as FPCk-1-1 cell line) into adenocarcinoma cells. Moreover, CRISPR/Cas9 technique is now well established, and the Authors should use it to KO/KD IL-17A receptor.

Minor:

Figure 1A: please change the filling colour of the body, since then in the graphs 1C, D... the non-FAP are white, while the FAP are yellow/brown.

Figure 1B: the first contour plot is not clear, so it is difficult to see from where the lymphocytes were gated. Please change it.

The choice of the lineage markers is interesting. Please justify it. Moreover, in the discussion, please add few sentences stating that it was not possible to analyse CD4+ ILC1, CD5+ mature ILC1 and immature ILCs, NKp80+ NK-like ILCs...

Supp Figure 1: I do not understand why ILC1 and ILC3 are negative for CD127 if they were gated as positive (according to the main Figure 1).

Line 247: ILC3 number should be changed with frequency.

Supp Figure 3B: I do not understand what the histograms on the right are quantifying, since on the left and from what the Authors claim is pretty clear that the frequency of colon NKp44+ ILC3 is increased. I am also misled by the colour code that is the one of the duodenum adenomas. The same colour code problem is present in figure 3 where the Authors are quantifying cells from the colon.

Figure 3B: "with the proportion of ILC3 among the overall colonic ILC pool being reduced in FAP adenoma". No statistical

analysis is provided to conclude this. The same for Figure 1F. The Authors should put standard error and statistical analysis in the bars.

Were the correlations shown in Figure 4 obtained with pool data from FAP and FAP adenomas? Since there are many dots, I guess this is the case. So, what happens to the correlations if the Authors split the 2 groups (even if one is small)? Despite the correlations with DDL1, the OP9DL1 to not induce changes in ILC3. The Authors should comment on this.

Reviewer #4

(Remarks to the Author)

The manuscript entitled 'IL-17A-Producing NKp44(-)ILC3s May Promote Duodenal Adenoma Formation in Familial Adenomatous Polyposis' by Kaiser and colleagues identifies an ILC population that is specifically enriched in the duodenum of FAP patients and is further increased in adenomatous duodenal samples of FAP patients. This particular population, NKp44(-)ILC3s, displays elevated IL-17A expression specifically in duodenal adenoma samples while not observed in colonic adenomas. The authors suggest that this population is involved in duodenal adenoma formation. Although the authors indeed demonstrate the presence of this particular population in duodenal adenomas, they do not provide any evidence for the involvement of this population in duodenal tumour formation. Due to complete lack of mechanistic insight this manuscript should not be eligible for Nature Communication in the current form. The following concerns were raised:

Major concerns

- No spatial validation of their findings. The entire manuscript is based on observations made using flow cytometry. The authors should demonstrate the presence of these specific populations in tissue samples. It currently remains unclear where these ILC3s are located in respect to the normal epithelium/adenomas, e.g. the mucosa/submucosa, near the crypt base/villus, top/base of the adenomas? In addition, by just using digested tissue biopsies it remains unclear what cells are being analysed by flow cytometry. For example, adenomatous tissue might also contain normal epithelium, and vice versa, normal epithelium might contain cells that have already lost the second APC allele or microadenomas that could not be detected macroscopically. The lack of spatial validation hampers the interpretation of the data and therefore prevents the identification of the mechanism by which IL-17A expressing cells might be involved in adenoma formation.
- In line with the previous point, the identification of upregulated genes in FAP adenomas has not been validated spatially using tissue sections. Are these genes upregulated in FAP adenomas derived from the epithelial compartment, and can they be pinpointed specifically to adenomatous regions or the normal epithelium surrounding it? In addition, it remains unclear why FAP adenomas are compared to non-FAP control and not FAP controls in figure 5b to identify markers specifically upregulated in FAP adenomas. As validation of the bulk RNAseq data the authors perform qPCR analysis, which raises the same concerns regarding biopsy digestion as the RNAseq and flow cytometry analysis.
- The authors suggest that ILC3s are involved in duodenal adenoma formation, however, by using tissues biopsies from established adenomas it remains unclear whether they have an active role in adenoma formation, or are just recruited to/enriched in established adenomas. If it has a role in adenoma formation, does IL17A addition to duodenal organoids from FAP patients drive oncogenic transformation? E.g. does it result in increased loss of APC in their in vitro cultures? Does it lead to morphological changes in the organoids? Does the cell type composition of the organoids change? Can they grow in the absence of Noggin and Wnt ligands? Again, no spatial/visual information regarding the organoid cultures is provided.
- In line with the abovementioned point, is this population also enriched in sporadic duodenal adenomas or is it FAP specific? And can it be detected in duodenal carcinomas (FAP or sporadic)? Could it have a role in tumour progression instead of adenoma formation?

Minor comments:

- The y-axis labels between the duodenal samples (figure 1) and the colon samples (figure 3) are different to indicate the same population, this is confusing.
- Some figures do not contain statistical (or n.s.) information, e.g. fig 2e,g,h or 3c.

Version 1:

Reviewer comments:

Reviewer #1

(Remarks to the Author)

The authors have addressed my concerns. The paper is convincing and advances the field

Reviewer #2

(Remarks to the Author)

The authors have addressed all reviewer concerns. This study provides novel insights into the status of innate lymphoid cells (ILCs) in human FAP. The authors report original findings on shifts in the ratios of ILC subsets, highlighting significant changes in ILC3 subsets and their expression of IL17. Notably, the increase in IL17 expression was more pronounced in

NKP44- ILC3s than in Th17 cells and was localized to the ileum rather than the colon. Furthermore, the authors identify NOTCH signaling from the tumor epithelium as a likely driver of IL17 expression.

The authors conducted RNAseq analysis to explore mechanisms linking the expansion of the NKP44- ILC3 subset to polyposis, providing further mechanistic insights. These findings shed light on the role of the local immune environment in the small bowel and colon in promoting tumor growth and potential crosstalk between these tissues via the Th17 response. This study opens a promising new avenue for future research into the role of ILCs and inter-tissue communication in the context of human cancers of small bowel and colon.

Reviewer #3

(Remarks to the Author)

I am satisfied with the new version of the manuscript.

Reviewer #4

(Remarks to the Author)

In the revised version of their manuscript, the authors have addressed almost all my concerns. Particularly the addition of spatial information enhances the understanding of their work. However, there is one critical point that was already raised in my previous comments that remains unanswered, which is the actual proof that ILC3-produced IL17A promotes duodenal adenoma formation (as the title states).

In their revised manuscript, the authors demonstrate that IL17A induces DUOX expression, ROS production and DNA damage in normal duodenal organoids from FAP patients. Although such ROS-high/damaged/inflammatory environment most likely could contribute to the development of many types of cancers it is unclear how this 'promotes' duodenal adenoma development. As the authors are most likely aware of, the initiation and development of adenomas in FAP is driven by loss of the second APC allele, causing hyperactivation of the Wnt pathway and resulting in increased proliferation and changes in differentiation patterns. The authors reveal that there is no difference in proliferation/differentiation in their duodenal organoids, and APC-loss has not been investigated in their manuscript (which can e.g. be demonstrated by culturing organoids in the absence of growth factors such as Wnt3a and Rspodin). Of note: APC gene expression is not a reliable method for determining APC-loss of function as this expression is highly dependent on the specific mutation in the APC gene. Instead assessment of Wnt target gene upregulation would be a better indication.

Given that the functional implications of ILC3-produced IL17A on APC loss and thus on adenoma initiation cannot be demonstrated in the current work, I would suggest to rephrase the passages in the manuscript indicating that 'it promotes adenoma formation' and instead discuss that their results indicate that these ILC3s could contribute to a local environment that makes the epithelium more submissive for oncogenic transformation (or something similar).

Minor:

- It would be helpful if the authors could add to their figures whether the data is of duodenal or colon origin (perhaps as a small sub-figure header or something). I see this is already done in some figures but not all, and this would definitely contribute to the readability/interpretation of the results as some of the figures throughout the manuscript look quite alike.
- Figure 3c the z-score legend is left completely blue (instead of Red-Yellow-Blue).
- Figure 5a, I think the y-axis of the correlation plot should be IL1B not IL23?
- Sup. Figure 2a y-axis IL-8 instead of 'IIL-8'

POINT-TO-POINT-RESPONSE TO THE REVIEWER COMMENTS

Reviewer #1 (Remarks to the Author): with expertise in ILCs

This study focused on investigating the role of intestinal innate lymphoid cells (ILCs) in familial adenomatous polyposis (FAP), an inherited gastrointestinal tumor syndrome associated with an increased risk of duodenal cancer. The researchers analyzed duodenal and colon samples from FAP and non-FAP patients using flow cytometry and RNA analysis. The findings revealed that FAP patients had a higher frequency of NKp44- ILC3s, in both normal and adenomatous duodenal tissue from these patients. These ILC3s were found to produce increased levels of IL-17A, and increased levels of IL-1 β , IL-23 and DLL4 in the adenomas may be involved in duodenal accumulation of IL-17A-producing ILC3s. The higher presence of this subset and IL-17A production correlated with elevated expression of certain genes related to inflammation and cancer development. Overall, the study suggests that IL-17A-producing NKp44- ILC3s may play a role in the formation of duodenal adenomas in FAP. While IL-17 production is commonly associated with NKp44- ILC3 cells, particularly during gut inflammatory circumstances, IL-17A expression under normal conditions has been rarely observed in adult tissues. Additionally, IL-17A is not produced by tonsil ILC3s, and the transcript is very low in adult ileum ILC3s. Thus, the increase of these population in the duodenum of FAP patients shows a uniqueness to this niche, which favors the development of these subsets. However, without a more comprehensive mechanistic understanding, it is challenging to definitively conclude that IL-1 β , IL-23, and DLL4 are directly responsible for the observed accumulation of IL-17A-producing ILC3s in FAP. Further information on the patients used in the study as well as experiments that employ more advanced techniques are warranted to establish a causal link and unravel the precise mechanisms driving this phenomenon.

Major points:

1 – The authors propose that increased levels of IL-1 β and IL-23, along with increased DLL4 expression, are involved in the duodenal accumulation of IL-17A ng, the paper falls short in providing a robust mechanistic understanding of this relationship. The study relies on correlational data and *in vitro* experiments, which are limited in elucidating the causal mechanisms at play. To strengthen their argument, it would have been beneficial for the authors to conduct functional studies that directly evaluate the impact of the niche on the accumulation of IL-17A-producing ILC3s in relevant *in vivo* models or *ex vivo* systems. For instance, the authors could co-culture with the organoids from FAP patients with CD34+ HSP or ILCPs, using the previously described protocols.

Thank you for your thoughtful comment on our study. We acknowledge the need for a deeper mechanistic understanding of how IL-1 β , IL-23, and DLL4 expression relate to the accumulation of IL-17A-producing ILC3s in FAP patients. While our study offers valuable correlational data and *in vitro* experiments, we agree that additional functional studies could strengthen these findings.

In response to your suggestion, we conducted further experiments, co-culturing FAP patient-derived organoids expressing DLL1 and DLL4 with ILCPs, supplemented with IL-1 β , IL-23, IL-2, and IL-7. This simulated niche environment

led to a marked increase in the NKp44(-) ILC3 phenotype and enhanced IL-17A production, as shown in Fig.5f and Supplementary Fig. 6D-F.

Notably, organoids alone did not produce ILC-specific cytokines like IL-23 and IL-1 β , highlighting the necessity of adding these cytokines (Supplementary Fig. 6F).

Additionally, we identified NKp44(-) ILC3s in FAP duodenum samples via MELC (Multi-Epitope Ligand Cartography) microscopy. These cells were primarily subepithelial but also in close contact with CD31(+) endothelial cells expressing DLL1 and DLL4, emphasizing the role of Notch ligands in generating NKp44(-) ILCs (Fig. 5E and Supplementary Fig. 6A).

The lines #484-492 and #499-506 include the respective changes in the manuscript.

We appreciate your suggestions, which have helped refine our study and advance the understanding of this field.

2 - Further information on the demographic data of the patients included in this study and their treatment status is warranted. The wide range of ages among the patients raises questions about potential age-related variations in the proportion of NKp44-ILC3s, especially given the previous observations that this population is reduced in adults. There is clearly a range of IL-17A production by these cells among the FAP patients, as well.

It would be valuable to present this information in a separate table, alongside the corresponding percentages of NKp44- ILC3s of CD45+ cells and % of IL-17A NKp44-ILC3s. Such data would enable a more thorough analysis of any potential correlations between patient demographics and the prevalence of this specific ILC3 subset and their function. It is also important to clarify whether the patients included in this study were receiving any specific treatments at the time of sample collection. Treatment regimens have the potential to impact the immune cell composition and function, thus potentially confounding the interpretation of the results.

We confirm that none of the patients included in this study were receiving any specific treatments at the time of sample collection, ensuring that treatment variables did not influence our results.

Regarding the age range, we acknowledge the relevance of age-related variations in NKp44(-) ILC3 populations, particularly the reduction in adults (Darboe et al., 2020). To address this, we have included a new table in Supplementary Data 1, which provides detailed demographic information, including patient ages, alongside the percentages of NKp44(-) ILC3s among CD45(+) cells and the proportion of IL-17A-producing NKp44(-) ILC3s for each patient.

Our analysis showed no significant differences in the proportion of NKp44(-) ILC3s or IL-17A production across different age groups (Figure for Review Only Fig. 1A and 1B). These findings suggest that age-related variations did not notably affect the prevalence or function of NKp44(-) ILC3s within our study population, reinforcing the robustness of our results.

We believe this additional data enhances the clarity of our work and supports a more detailed analysis of potential correlations between patient demographics and the ILC3 subset under investigation.

3 – The authors primarily emphasize the production of IL-17A by NKp44- ILC3s, yet it is essential to consider the potential involvement of Th17 cells, which are also known to produce IL-17A. It would be important for the authors to investigate whether Th17 cells are increased and whether they exhibit higher IL-17A production.

We agree that this is an important issue that was not adequately addressed in the initial version of our manuscript. In response, we have examined Th17 cells and their IL-17A production in our revised manuscript.

Our data indicate no significant differences in the frequencies of CD4(+) T cells between non-FAP, FAP, and FAP adenoma tissue (Figure for Review only 2A). Additionally, the frequencies of IL-17A(+) CD4(+) cells among CD45(+) lymphocytes was similar across groups following P/I stimulation (Fig. 3B).

Importantly, immunohistochemistry of FAP adenomas showed an increased presence of IL-17A cells, which were predominantly CD3-negative, indicating that other cell types, such as ILC3s, are the primary source of IL-17A in these tissues (Fig. 3A). These data suggest that Th17 cells may have a less prominent role in IL-17A production in FAP. Lines #419-431 have been updated to reflect the changes outlined in the manuscript.

4 – Although the authors show that blocking IL17A in the culture supernatants reduced the upregulation of DUOX2 and DUOXA2, it would be relevant to further understand the function of these cells. For instance, similar to the experiment in Figure 5, the authors could sort NCR+ and NCR- ILC3s separately from patient samples and check for DEGs between these subsets. This analysis could also shed light into potential crosstalk between NCR- ILC3s with other cell types as well as to a more in-depth phenotyping of NKp44- ILC3s.

We appreciate your suggestion regarding further investigation of NKp44(+) and NKp44(-) ILC3 subsets. In response, we sorted NKp44(+) and NKp44(-) ILC3s from patient samples and conducted bulk RNA sequencing. Our analysis revealed distinct sets of differentially expressed genes (DEGs) between the two populations. Enrichment analysis showed significant upregulation of genes in NKp44(-) ILC3s related to the “KEGG pathway for Cell Adhesion Molecules (CAMs)” and “GOMF chemokine receptor binding” (Fig. 2F). These findings align with our flow cytometric analysis of IL-17A-producing NKp44(-) ILC3s, which displayed high expression of epithelial retention markers such as α E β 7 integrin in adenomas (Fig. 3C and 3D).

This suggests that NKp44(-) ILC3s are not only present but actively retained in the tissue, potentially through interactions with epithelial and stromal cells. The upregulation of adhesion and retention markers indicates that these cells are positioned to play a prolonged role within the tissue microenvironment, potentially contributing to local immune regulation and adenoma development. The respective modifications have been implemented in lines #413-418 and #440-446 of the manuscript.

5 – In addition, the authors show the surface markers expressed by the ILC3 cluster but fail to show how the expression of each marker as well as the cytokine production differs between each of the ILC3 subsets. For instance, are the IL-17A+ NCR- ILC3s

preferentially CD56 negative, as previously reported?

We agree that a detailed analysis of surface marker expression and cytokine production between ILC3 subsets is crucial to better understand their functional differences. In the revised manuscript (Fig. 3C, Supplementary Fig. 4A-C), we have extended this analysis.

We now provide detailed expression profiles of several surface markers, including CD56, CD103, IL1R1, HLA-DR, CD200R1 and CD49a, as well as homing markers such as integrin β 7, CCR6, CCR9, CXCR6 and CXCR3. We also examined transcription factors (c-Maf, T-bet, ROR γ t) and the proliferation marker Ki-67, comparing NKp44(-) and NKp44(+) ILC3s. We also assessed these markers on NKp44(-) ILC3s in relation to IL-17A production.

Our findings indicate that CD56 is expressed on both NKp44(-) and NKp44(+) ILC3s, with a higher expression on IL-17A-producing NKp44(-) ILC3s. With regard to transcription factors, we observed only minimal differences between IL-17A(+) and IL-17A(-) cells. Of note, mucosal retention markers like α 4 β 7 and α E β 7 (intraepithelial specific) as well as CCR6 were predominantly expressed on IL-17A(+) NKp44(-) cells, with a similar profile seen in NKp44(+)IL-17A(+) cells. These results are presented in the revised manuscript as a heatmap of expression levels (z-scores) in the new Fig. 3C and Supplemental Fig. 4C, offering a more detailed characterization of these subgroups and their potential differences. We have included the requested changes in the manuscript, specifically in lines #410-413 and #432-450.

6 - The authors report that stimulation of duodenal organoids with IL-17A specifically induced upregulation of DUOX2 and DUOXA2, while no significant effects observed for other studied genes. Although this observation provides insight into the potential role of IL-17A in the regulation of DUOX2 and DUOXA2, the limited analysis of gene expression leaves important questions unanswered. To gain a more comprehensive understanding of the transcriptomic changes induced by IL-17A, it would be beneficial for the authors to perform bulk RNA-seq analysis of the organoids after exposure to IL-17 or the supernatant from NCR- ILC3s culture. If the latter is chosen, it would be crucial to stimulate the organoids with the supernatant of other ILC cell culture, such as NKp44+ ILC3s, not just media as in Figure 7B. This could reveal additional genes or pathways influenced by IL-17A signaling in the context of duodenal adenoma formation in FAP.

We agree that a more comprehensive analysis of transcriptomic changes induced by IL-17A in duodenal organoids is essential for a deeper understanding. Based on your suggestion, we conducted additional experiments.

We stimulated duodenal organoids from both FAP and non-FAP patients with IL-17A and conducted bulk RNA-seq analysis. The results, now incorporated in Fig. 7E-G, highlight differentially expressed genes (DEGs) when comparing IL-17A-stimulated organoids to unstimulated controls.

Pathway and enrichment analyses of DEGs revealed several enriched pathways increased in FAP patients compared to non-FAP patients. Notably, we observed enrichment in the “GOMF Signaling Receptor Regulator Activity” pathway, suggesting IL-17A may modulate receptor activity and cellular responses. The “GOBP Regulation of Hydrogen Peroxide Metabolic Process” pathway was also enriched, supporting the role of IL-17A in oxidative stress regulation, consistent with DUOX2 and DUOXA2 function.

Furthermore, DUOXA2 was significantly upregulated in FAP-derived organoids, whereas non-FAP organoids showed no statistical significance in IL-17A-induced changes (Supplementary Data 2). In line with the role of DUOX2 in ROS production, we demonstrated that IL-17A increased Duox2 protein levels, ROS production, and DNA damage (γ H2AX) in FAP-derived organoids, independent of mitochondrial ROS as indicated by MITOSOX (indirect mitochondrial ROS detection) staining (Fig. 8F-I and Supplementary Fig. 8B and 8C). The revisions are visible in lines #558-568, #580-596 and for the discussion part in lines #687-694 .

Recent publications by other groups (<https://doi.org/10.1016/j.ccell.2023.12.006>, <https://doi.org/10.1002/mco2.524>, <https://doi.org/10.1016/j.freeradbiomed.2023.06.012>, doi: 10.21203/rs.3.rs-3406046/v1) corroborate these findings, demonstrating the induction of DUOX2 by IL-17A and linking this to increased ROS production in the context of tumorigenesis.

We have updated our manuscript in lines #672-676 to include these references.

We believe this additional data provides a more thorough understanding of IL-17A's role in duodenal adenoma formation in FAP, addressing the important points you raised.

Minor points

1 – In Supplementary 2B, correct CD200R for CD200R1 instead.

We apologize for this mistake and have corrected the receptor name to CD200R1 in the respective figures accordingly.

2 - In Supplementary 2B, CD117 is higher on normal tissue and decreases in FAP. The authors should clarify how was ILC3s gated on the samples when CD117 was lower. Do they have additional markers to confirm they were gating on ILC3s?

We apologize for any confusion caused by the presentation of CD117 expression data. Upon further analysis, we confirm that there is no significant difference in CD117 expression between NKp44(-) and NKp44(+) ILC3s across the groups (Figure for review only 2B).

To clarify, while CD117 is a key marker, we also used additional markers such as ROR γ t and CD161, along with the observed cytokine expression patterns, to reliably identify ILC3s (Supplementary Fig. 2C). These markers provide robust confirmation that we were gating on ILC3 populations, despite any variations in CD117 expression. We hope this addresses the concern and clarifies our approach in identifying ILC3s.

Reviewer #2 (Remarks to the Author): with expertise in CRC, intestinal immunology

Kaiser et al report on the role of ILCs in duodenal adenomas. The authors report collection of normal duodenal tissue from FAP patients (n=82, 40/82 female, age 17-84) and non-FAP patients (n=26, 12/26 female, age 15-72), as well as duodenal polyps from patients (n=16, 10/16 female age 16-72). Furthermore, they collect control samples: normal colon tissue from FAP patients (n=26, 14/26 female, age 18-67) and non-FAP controls (n=11, 6/11 female, age 24-58), colonic adenomas from FAP patients (n=14, 7/14 female, age 19-43). To study ILCs in the lesions, they enzymatically dissociate adenoma biopsies, isolate mononuclear cells, and analyze them by FACS using an extensive dump gate, and differentiating the three major types of ILCs (1, 2 & 3) base on expression of CD127 (IL-7 receptor) and CD117 (cKIT; ILC3) and CRTH2 (Prostaglandin DP2 receptor; ILC2). They distinguish t two subsets of ILC3s as NKp44 (Natural Killer Cell Receptor) negative or positive based on their expression of NKp44. They find increased densities of ILC1s and ILC3s but not ILC2s in the FAP adenomas, relative to healthy tissue from FAP or non-FAP patients. Among ILC3 the increase was restricted to the NKp44 negative population. The show that a higher frequency NKp44 negative population produces more IL17a in the adenomas but find no difference for NKp44 positive population of the ILCs. Also, expression of IL17 was restricted to the duodenum polyp ILCs and was not obvious among the colon polyp ILCs. Interestingly, the vast majority of healthy colon ILC3s were NKp44 positive, and overall, the colon had more ILCs than the small bowel. Using bulk RNAseq they observed increased expression of IL1b, IL23, and the Notch ligands Delta-like 1 and 4 (DLL1 and DLL4) in duodenal polyps. They also demonstrate a correlation with increased expression of IL17A and IL23 or DL1 or DL4. To better understand the relation between these the culture NKp44 negative ILCs in the presence of IL23 and IL1b and show that addition of dl4 increases expression of IL17a by the cells. Stimulation of the organoids with IL17a induced expression of DUOX2 (dual oxidase 2). DUOX2 expression by polyp organoids was upregulated in response to IL17a or co-culture with ILC3s. The authors conclude that the ILC3s have a role in the growth of duodenal adenomas. This is the first study of ILCs in human duodenal adenomas and adds substantially to our knowledge of the role of ILCs in small bowel cancer. The findings are not completely in line with an earlier study by Sonnenberg and colleagues (PMID: 34407392) which reported down regulation of ILC3s in tumors of CRC patients. The authors do not quote this article. Also, the findings contrast with similar studies done in mice and are therefore somewhat surprising. The authors report that the NCR- rather than the NCR+ ILC3s expand in polyps and have proinflammatory properties. This contrasts with the proposed Treg-inducing role of NCR- ILC3s and the proinflammatory potential of the NCR+Tbet+ ILC3s in mice. Additional work is needed to establish

whether this difference between human and mouse ILC3s is real. The work is done very carefully and with rigor. However, the conclusion that ILC3s contribute to polyp growth is premature and not strongly supported by the provided data. The story would have been complete had the authors shown that the NCR- ILC3s or supernatant of these cells enhance polyp growth and/or invasive properties. This could be done using an *ex vivo* assay or better also using xenogeneic tumor transplant in immune-deficient mice. As it stands the manuscript is of high interest, but additional data is needed to validate the conclusions.

Thank you for your positive feedback on our study and for recognizing its significance. We appreciate your detailed and constructive comments, and we would like to address the points raised in a structured manner.

1. Citing Sonnenberg et al.:

Thank you for highlighting the study by Sonnenberg et al. We have now included this citation in our revised manuscript (citation No. 20, line #142). While their work focused on ILC3 downregulation in CRC patients, our study centers on ILCs in human duodenal normal mucosa and adenomas, providing insight into small bowel cancer. The differences between our findings and theirs may be attributed to the distinct microenvironments of the duodenum versus the colon, which likely contribute to the varying roles of ILCs across different regions of the gastrointestinal tract. Additionally, our data on colonic adenomas in FAP patients (Fig. 4A-C) also show downregulation of NKp44(+) ILC3s, which aligns with their findings. The variability of ILC distribution in the gastrointestinal tract has been previously reported by our group (<https://doi.org/10.1371/journal.ppat.1006373>) and is also shown in Supplementary Fig. 5A-B, supporting the idea of compartment-specific effects.

1. Differences Between Mouse and Human ILC Models:

We agree that the observed differences between our study and mouse models are intriguing. Species-specific variations and distinct cellular dynamics likely explain these discrepancies. In mouse models, the expansion of NCR+ Tbet+ ILC3s with proinflammatory properties contrasts with our findings in human adenomas, where NCR(-) ILC3s appear to play a protumorigenic role. This discrepancy may reflect differences in the tumor microenvironment between species and warrants further investigation. While it would be interesting to investigate the effect of ILC3s in a xenogeneic tumor model, we did not have access to such a model, and initiating an animal study would require extensive approvals, which were beyond the scope of this project and unlikely to be obtained within a reasonable time frame. Furthermore, it is uncertain whether the duodenal microenvironment in a mouse model would accurately replicate the conditions in FAP patients, which is why we focused on the *ex vivo* model instead.

2. Ex Vivo Assays and Mechanistic Insights into ILC3 Function:

Although we were unable to keep the duodenal organoids in long-term culture to directly demonstrate adenoma formation, we performed additional experiments to provide mechanistic insights. In our study, we demonstrated that FAP-derived organoids can induce IL-17A-producing NKp44(-) ILC3s, and that IL-17A leads to increased ROS production and DNA damage, evidenced by elevated γ H2AX levels in these organoids (Fig.5F, Fig. 8F-I and Supplementary Fig. 8B). This supports our hypothesis that NKp44(-) ILC3s may contribute to adenoma formation via IL-17A-mediated oxidative stress, providing a mechanistic link between IL-17A production and tumorigenesis (lines #580-596 and #687-694).

3. Spatial Analysis and ILC3 Marker Expression:

We have strengthened our hypothesis with additional spatial analysis showing an enrichment of IL-17A(+) CD3(-) cells in adenomas via classical immunohistochemistry (Fig.3A). Using the advanced MELC microscopy technique, we further identified ILC3s within the tissue, although cytokine detection was not possible with this method (Fig.1F and Supplementary Fig.3). Nevertheless, flow cytometry confirmed that NKp44(-) ILC3s are more abundant in FAP adenomas. Additionally, our organoid co-culture experiments demonstrated that IL-17A can induce DUOX2 expression, increase ROS production, and cause DNA damage in FAP-derived organoids (Fig.5F, Fig. 8F-I and Supplementary Fig. 8B). These data collectively suggest that NKp44(-) ILC3s play a role in adenoma formation through their proinflammatory properties (lines #419-424, #687-694 and #580-596 and #687-694).

We recognize that direct evidence showing that NKp44(-) ILC3s or their supernatant enhance polyp growth via *ex vivo* assays or xenograft models is currently lacking. However, our data indicate that IL-17A-producing NKp44(-) ILC3s promote key processes such as oxidative stress and DNA damage in duodenal organoids, supporting their potential role in adenoma formation. These findings, although not demonstrating direct tumor growth, provide a strong mechanistic basis for further investigation into the role of ILC3s in human duodenal adenomas.

In conclusion, we sincerely appreciate your valuable feedback, which has helped us improve the depth and clarity of our manuscript. We hope the additional data and explanations we have provided address your concerns and contribute to a more comprehensive understanding of the role of ILC3s in duodenal adenoma formation.

Reviewer #3 (Remarks to the Author): with expertise in ILCs

In this manuscript Kaiser KM et al., claim that their data indicate a role for IL17A secreting NKp44- ILC3 in duodenal adenoma formation in FAP, but I do not think that the results support this conclusion. The only 2 differences that the Authors show between normal FAP and adenoma FAP is that ILC3 are increased and that IL17A+ILC3 are increased but only if considered as percentage of CD45+ cells. I think that what the Authors show is that IL17A+ILC3 are associated with FAP in general.

Major:

Since CD45+ cells do not change, is the ILC1 and ILC3 increase due to proliferation? Or they were recruited in the intestine? How is the expression of gut homing receptors in these 2 populations? Are the ILC1 functional?

Thank you for your helpful comments and suggestions.

We apologize if the impression was given that the only notable differences between normal FAP and FAP adenomas were the increases in ILC3s and IL-17A(+) ILC3s, particularly when considered as percentages of CD45(+) cells. We would like to clarify this (see also answer to next question) and provide more detailed responses to your specific questions.

1. Proliferation vs. Recruitment:

To address your concern regarding whether the increase in ILC1 and ILC3 populations is due to proliferation or recruitment, our Ki-67 data suggest that proliferation is not a major factor. Ki-67 expression was not significantly different between FAP adenoma tissue and normal FAP or non-FAP controls, indicating that the observed increases in ILC populations are unlikely to be driven by local proliferation (Figure for Review only 3).

2. Intestinal Recruitment and Retention:

In terms of recruitment, we observed a modest increase in CCR9 expression, particularly in FAP adenomas, suggesting that enhanced recruitment may be possible. However, CCR9 expression was only found on a subset of ILC3s, and there were no significant differences in CCR9 expression between IL-17A(+) and IL-17A(-) ILC3s (Supplementary Fig. 4A). This suggests that while recruitment through CCR9-mediated homing may occur, it likely plays a limited role in driving the observed increase in IL-17A(+) ILC3s.

Regarding other chemokine receptors analyzed, we found no significant differences in their expression between the groups, further supporting the conclusion that recruitment is not the primary mechanism.

The most notable factor contributing to the increased presence of IL-17A(+) NKp44(-) ILC3s in FAP adenomas appears to be enhanced retention. Our integrin expression data show that IL-17A(+) NKp44(-) ILC3s exhibit high levels of β 7-integrin, particularly α E β 7, which promotes epithelial retention (Fig. 3C/D). This retention mechanism is further supported by our

immunohistochemistry (IHC) and MELC microscopy data, which show a predominant localization of IL-17A(+) CD3(-) cells in the epithelial regions of FAP adenomas (Fig. 1F and 3A).

Additionally, our bulk RNA-seq data reinforce this observation, with enrichment analysis revealing significant upregulation of the "KEGG Cell Adhesion Molecules (CAMs)" pathway in NKp44(-) ILC3s compared to NKp44(+) ILC3s in FAP adenomas (Fig. 2F). This underscores the role of integrin-mediated retention and adhesion in the accumulation of these cells in the adenoma microenvironment.

Furthermore, we observed an alteration of the local microenvironment within FAP adenomas, with increased expression of IL-23, IL-1 β , and the Notch ligand DLL4. These factors, as demonstrated by our *in vitro* experiments, strongly support the development of IL-17A-producing NKp44(-) ILC3s. This suggests that the local cytokine and signaling milieu not only fosters the differentiation and expansion of these cells but also reinforces their functional proinflammatory phenotype.

In summary, our findings indicate that the increase in IL-17A(+) NKp44(-) ILC3s in FAP adenomas is primarily driven by a combination of local retention and the supportive microenvironment. Proliferation is unlikely to play a major role, as indicated by our Ki-67 data, and while recruitment via CCR6 and CCR9 may contribute to a limited extent, it is not the predominant mechanism. Instead, the elevated levels of *IL-23*, *IL1B*, and *DLL4* in the adenoma microenvironment support the differentiation and persistence of IL-17A-producing NKp44(-) ILC3s, which are retained in the epithelial region via integrin-mediated adhesion. This suggests that FAP adenomas create a niche that actively promotes both the development and the retention of these proinflammatory cells, contributing to adenoma formation. The corresponding sections of the manuscript have been updated (lines #401-450, #484-492, and #499-506).

3. ILC1 Functionality:

Regarding the functionality of ILC1s, our flow cytometry data confirm that ILC1s are indeed functional in FAP adenomas. While their characteristics differ significantly from those of NKp44(-) ILC3s, they remain an active immune population in these tissues (Figure for Review only 2C).

Figure 2B and E: The Authors state that IL17A production was most prominent in adenoma tissue. However, there is no difference between FAP and FAP adenoma. Thus, the IL17A increase could be a shared phenomenon of FAP.

Thank you for your careful review of Fig. 2B and E, and we apologize for the confusion caused by the initial presentation of the data. We appreciate your observation and would like to clarify the matter.

It was an oversight on our part not to indicate the significance values in the initial figure. Upon reevaluation, we have now included the appropriate significance annotations in the revised version of the manuscript. The difference in IL-17A production between FAP normal tissue and FAP adenomas is, in fact, significant, and this is now clearly marked in the updated figures (Fig. 2B/D).

That being said, we also agree with your broader point that the increase in IL-17A production is indeed a general feature of FAP. Our data show that even in FAP normal tissues, there is an elevated frequency of IL-17A-producing ILC3s compared to control non-FAP tissues, which suggests that IL-17A production is a characteristic phenomenon in FAP, not just restricted to adenomas. This shared increase in both FAP normal and adenoma tissues reflects the underlying inflammatory environment in FAP, which could contribute to adenoma development.

We have revised our manuscript to better highlight this point, emphasizing that while IL-17A production is more pronounced in adenomas, it is already elevated in FAP normal tissues compared to non-FAP controls (Fig. 2B/D and Fig. 3A). The updates are now in lines #432-437.

The Authors claim that there is no difference in phenotype, however in the Supp Figure 2 CD56 seems upregulated in FAP adenoma. How is the quantification? How is the expression of NKp46? Also, ILC3 were characterized as cells expressing CCR6 but negative for CXCR3. Are the ILC3 identified in this way different in frequency between FAP and FAP adenoma? CCR6+NKp46+ ILC3 are a source of IL-22 and IL-17A. The Authors should evaluate this population. Also, how is the expression of RORgt, T-bet and c-Maf? Are these cells more or less prone to become proinflammatory ILCs producing IFN-g?

Thank you for your thoughtful questions, which allow us to clarify and expand upon our findings.

We understand that our initial presentation may have given the impression that there were differences in CD56 expression between the groups. However, upon further analysis, we can confirm that no significant differences in CD56 expression were found between FAP normal and FAP adenoma tissues on NKp44(-) or NKp44(+) ILC3 subsets (Supplementary Fig. 2B and “Resource Data” with statistical tests).

Regarding NKp46 expression, we observed that it is generally low to absent in NKp44(-) ILC3s across both FAP normal and FAP adenoma tissues. While NKp44(+) ILC3s occasionally exhibit higher NKp46 expression, these instances are infrequent and do not represent a substantial portion of the ILC3 population (Figure for Review only 4A).

Both NKp44(-) and NKp44(+) ILC3 subsets are primarily CCR6(+) and CXCR3(-), with no significant differences in frequency between FAP normal and FAP adenoma tissues (Supplementary Fig. 2C). Thus, the CCR6+CXCR3(-) ILC3 population is consistent across both tissue types without significant variation in frequency.

We also examined the CCR6+NKp46+ ILC3 subset, which is known to be a source of IL-22 and IL-17A. In our analysis, this population is present at very low levels and is nearly absent among NKp44(-) ILC3s, particularly in FAP adenoma tissues (Figure for Review Only 4A-B). This suggests that while these cells may contribute to IL-17A and IL-22 production in other contexts, their role in FAP adenomas appears limited.

In terms of transcription factors, we observed relatively low levels of ROR γ t and c-Maf expression, with no significant differences between NKp44(-) and NKp44(+) ILC3 subsets (Supplementary Fig. 2C). T-bet expression was also low in both populations. The ILC3 subsets analyzed were predominantly GATA3(-) and T-bet(-), indicating that these cells are less likely to adopt a proinflammatory, Th1-like phenotype under normal conditions.

Although we did not observe IFN γ production after immediate PMA/Ionomycin stimulation, we found that under specific conditions, such as prolonged stimulation with IL-1 β , IL-23, IL-2, and IL-7 on OP9-DL4 feeder cells, these ILC3s are capable of producing IFN γ (Figure for Review only 4C). This demonstrates that while they do not show a proinflammatory phenotype *in situ*, they have the potential to become IFN γ -producing ILCs under the appropriate environmental conditions.

In summary, we have carefully reexamined the phenotypic and functional characteristics of ILC3s in FAP tissues. Although certain subsets, such as CCR6(+)NKp46+ ILC3s, are present at low levels, the ILC3 population in FAP adenomas predominantly exhibits a CCR6(+)CXCR3(-) phenotype with limited proinflammatory activity under steady-state conditions. However, these cells do have the capacity to become proinflammatory when exposed to the appropriate cytokine environment, as demonstrated in our *in vitro* experiments. In lines #438-446 of the manuscript, the respective phenotypic characteristics of the subsets were presented.

OP9 experiment: I think that this experiment is not designed in the correct way (and the methods do not match with what is written in the results). OP9 stromal cells are used as feeders either to differentiate ILCs from CD34+ precursors, or to generate ILC clones. In this work, according to what is written in the results, NKp44- ILC3 (bulk?) were incubated for 3 days with cytokines and different OP9. First, if the Authors wanted to show the impact of Notch ligands, they could have added recombinant proteins expressing only the extracellular portion of Dll1 or 4, instead than the OP9. Second, in the first column w/o ILC3 which are the cells producing IL17A, the OP9, OP9DL1 or OP9DL4? IL17A secretion in the presence of only IL1b and IL23 (or with medium alone) is a control that need to be shown. In any case, in my opinion, the Authors should do single cell cloning with ILC3 obtained from non-FAP, FAP and FAP adenoma using the different OP9 to compare the percentages of clone formation and the identity of the clones. To confirm the identity of the clones ICS including transcription factors

and type1, 2 and 3 cytokines should be used.

We apologise for the confusion caused by the original version of Fig. 4I. The purpose of this figure was to show that only in the presence of OP9 cells expressing DLL4 was there a significant induction of IL-17A secretion. In this experiment we tested ILC3s cultured alone or in co-culture with OP9 cells, OP9-DL1 or OP9-DL4 cells. We acknowledge that our initial labelling of ILC3 cultures without OP9 cells as "w/o ILC3" was misleading and may have caused confusion. The labelling implied that "w/o" referred to the absence of ILC3s, which was not the case. We have corrected this in the revised Fig. 5F. In addition, following your suggestion, we now show that OP9-DL4 cells alone, with or without cytokine stimulation, are not able to produce significant levels of IL-17A, which supports our original findings.

Regarding the use of soluble Notch ligands, we performed additional experiments with soluble DLL4 (sDLL4) to investigate whether it could mimic the effects of membrane-bound DLL4. However, we did not observe an increase in the number of NKp44(-) ILC3 cells in these experiments (Figure for Review only 4D). This finding suggests that membrane-bound Notch ligands may provide a more effective interaction than their soluble forms, as reported in other studies (doi: 10.1038/onc.2008.229). In addition, we suspect that NKp44(-) ILC3s may require other microenvironmental factors for survival and differentiation over time, as suggested by previous research (doi: 10.1089/ten.teb.2014.0547).

We also understand your concern about the control experiments in our original design. You raised a valid point regarding the need to show IL-17A secretion in the presence of cytokines alone or with OP9-DL4 cells without ILC3s as controls. We have now included these controls in the revised manuscript to provide a more thorough and balanced analysis (Fig. 5F).

Your suggestion to explore single cell cloning with ILC3s from non-FAP, FAP and FAP adenoma tissues using different OP9 stromal cell lines is an interesting idea. While we agree that single cell cloning could provide valuable insights into the clonal diversity and identity of these cells, we believe this is beyond the scope of the current study. However, to further support our findings, we performed additional bulk culture experiments. We co-cultured ILCPs with duodenal organoids (expressing DLL4 as shown in Supplementary Fig. 6C) and demonstrated that only the combination of ILCP co-culture with organoids and cytokines led to the significant generation of IL-17A-producing NKp44(-) ILC3s (Supplementary Fig. 6D-F). This supports our hypothesis that both DLL4 and cytokines are required for the differentiation of this ILC subset.

In summary, we have carefully revised our manuscript to address these points and provide a more comprehensive analysis (lines #499-506 in the Results section and #656-663 in the Discussion). We appreciate your thoughtful feedback, which has helped to improve the clarity and rigour of our study.

In patients, which cells are expressing DLL1 or DLL4?

Using MELC (Multi-Epitope-Ligand-Cartography) microscopy we observed that DLL1 and DLL4 are co-expressed with CD31 and epithelial cells, indicating their presence on both the apical side of epithelial cells and on vascular endothelial cells (Fig. 5E and Supplementary Fig. 6A).

Furthermore, we demonstrated that NKp44(-) ILC3 cells are located within the epithelial region and near CD31+ endothelial cells, suggesting potential contact and influence between these cells (lines #484-492 in the Results section and #656-658 in the Discussion).

RNAseq: The Authors show only the difference between FAP adenoma and control. What is the difference between FAP and FAP adenomas?

We acknowledge your inquiry regarding the transcriptional differences between FAP normal tissues and FAP adenomas. To address this, we expanded our analysis, now presented in Fig. 6.

Our analysis identified 112 upregulated and 48 downregulated genes when comparing FAP adenomas to normal FAP tissue (fold change > |1.5|, adjusted p-value < 0.05; Fig. 5B, Supplementary Data 2). When comparing FAP adenomas to non-FAP control tissue, we identified 161 upregulated and 84 downregulated genes. Notably, 103 genes were differentially expressed in both comparisons, with 76 upregulated and 27 downregulated. This overlap suggests that gene expression changes associated with adenoma formation are already present in FAP normal tissue, indicating that even in the absence of visible adenomas, the normal mucosa in FAP patients shows molecular alterations linked to adenoma development (Fig. 6C, Supplementary Data 2).

This expanded analysis has been included in the manuscript to provide a clearer understanding of the gene expression changes in FAP adenomas, as well as the early molecular alterations seen in normal FAP tissue (lines #520-524 in the Results section).

The Authors only focused on IL-17A, but in the last experiment with the IL-17A blocking antibody it is clear this is not the only player. What about IL-17F for instance?

We agree with your observation regarding Fig. 7C, where DUOX2 and DUOX2A2 expression remained elevated in organoids cultured with ILC3 supernatants, even after IL-17A blockade, compared to unstimulated organoids. This indeed suggests that IL-17A might not be the only factor driving DUOX2/DUOX2A2 expression. Alternatively, incomplete IL-17A neutralization due to insufficient antibody titers might also be a factor. Furthermore, we noted that only a small subset of unstimulated organoid cultures showed lower expression compared to organoids after IL-17A blockade.

To address your suggestion, we conducted additional experiments with an anti-IL-17F antibody, while also repeating the entire experiment with all conditions to

avoid bias. The results showed that blocking IL-17A effectively prevented the supernatant-induced upregulation of DUOX2 and DUOXA2, though expression levels did not return fully to those of unstimulated organoids. In contrast, blocking IL-17F did not inhibit the upregulation of these genes, indicating that IL-17F does not play a significant role in this context.

These findings confirm that IL-17A is a major driver of DUOX2/DUOXA2 expression, while IL-17F appears to have little effect. The revised manuscript in lines #579-580 and supplementary materials now reflect these additional findings (Supplementary Fig. 8A).

The fact that ILC3 supernatant induces DUOX2 or its active metabolite DUOXA2, which as per the Authors “are thought to promote cancer cell appearance and progression” does not prove that they actually do it. Moreover, only one gene (and an associated one) is not indicative of any change.

We agree with your point that the upregulation of a single gene or even a pair of associated genes is unlikely to fully explain the complexity of adenoma formation. Our bulk RNA-seq data from FAP adenoma and FAP normal tissues show that a number of genes associated with tumour development are significantly upregulated compared to control tissues. While we do not claim that ILC3s and their production of IL-17A are the sole contributors to this process, our findings consistently suggest that this population plays a relevant role in FAP-associated duodenal adenoma development.

To further address your concerns, we performed additional experiments and analyses. Specifically, we stimulated organoids derived from both FAP and non-FAP tissues with IL-17A and performed bulk RNA sequencing. Our results showed that both DUOX2 and DUOXA2 were differentially expressed in FAP-derived organoids, whereas DUOXA2 was not significantly upregulated in non-FAP organoids. Subsequent pathway enrichment analysis revealed the activation of several pathways, including those related to IL-17 signalling, in both FAP and non-FAP organoids (Fig. 7E and Supplementary Data 2). Notably, FAP-derived organoids showed a specific enrichment in the GO pathway 'Regulation of Hydrogen Peroxide Metabolic Process', suggesting that IL-17A may affect oxidative stress mechanisms more prominently in FAP cells (Fig. 7G). This is consistent with the observed upregulation of DUOX2/DUOXA2 in FAP tissues and the role of oxidative stress in adenoma formation/tumourigenesis.

Furthermore, at the protein level, we confirmed that IL-17A stimulates Duox2 production, which is associated with increased ROS levels (Fig.8F and Supplementary Fig. 8B). Interestingly, the use of MITOSOX - a dye specific for the detection of mitochondrial superoxide - did not show an increase in mitochondrial ROS (Supplementary Fig. 8C). This suggests a mitochondria-independent mechanism, probably mediated by DUOX2/DUOXA2. In addition, we observed increased DNA damage, as evidenced by increased γ H2AX staining, following IL-17A stimulation in FAP organoids (Fig. 8G-I and Supplementary Fig. 8B).

Taken together, our results support the role of the IL-17A → DUOX2 → DNA damage cascade in driving tumourigenesis, particularly in the context of FAP. This mechanism is also supported by accumulating evidence from the literature (e.g. <https://doi.org/10.1016/j.ccell.2023.12.006>, <https://doi.org/10.1002/mco2.524>, <https://doi.org/10.1016/j.freeradbiomed.2023.06.012>, DOI: 10.1097/HC9.000000000000454), which further strengthens our conclusions.

These updated results are now reflected in the lines #558-568 & #580-612 in the Results section and #672-676 & #687-694 in the Discussion and are illustrated in Fig.7&8 and Supplementary Fig. 7&8.

How are these cells behaving in tumour invasion assays?

In a separate project, we tested the tumor invasion potential of human ILC1 cells in hepatocellular carcinoma using a chorioallantoic membrane (CAM) assay. From that experience, we learned that very high cell numbers are required to accurately track the cells within the tumor tissue. Unfortunately, for NKp44(-) ILC3 cells, we were unable to isolate enough cells from our FAP samples to conduct a similar assay.

A tumor invasion assay would certainly be an interesting approach, but we believe it may not provide essential information for our specific project. Given the nature of our study, the most relevant insights come from our immunofluorescence (IF) and MELC microscopy data, which clearly show the accumulation of NKp44(-) ILC3s in FAP adenomas. These techniques allow us to directly visualize their presence in the tumor microenvironment, which we consider crucial for understanding their role in this context.

Thank you for your understanding, and we appreciate your thoughtful suggestions to help improve our study.

Do they migrate/proliferate more?

Our flow cytometry analysis did not reveal increased Ki-67 expression in NKp44(-) ILC3 cells from FAP adenoma tissue (Fig. 4A), nor did *in situ* analysis show co-expression of Ki-67 with NKp44(-) ILC3 cells (Supplementary Fig. 3). This suggests that increased proliferation is unlikely to be the main driver of NKp44(-) ILC3 accumulation in FAP adenomas.

Regarding migration, we did not observe significant differences in the expression of key chemokine receptors between NKp44(-) ILC3 populations in FAP normal and adenoma tissues. While enhanced migration or recruitment seems unlikely to be the primary mechanism, it is still possible that some level of migration occurs, with circulating NKp44(-) ILC3 or ILC precursors entering the duodenum and differentiating into IL-17A-producing cells under the influence of the local microenvironment.

That being said, our integrin expression data and bulk RNA-seq analyses point towards retention as a key factor in the accumulation of NKp44(-) ILC3s at the site. The increased expression of integrins such as $\alpha E\beta 7$, along with the upregulation of adhesion-related pathways, suggests that retention in the local tissue is of particular importance for their presence in the adenoma.

The respective modifications have been implemented in lines #438-446 of the manuscript to incorporate this relevant question.

Thank you for your consideration and understanding.

It would be important to have specimens from duodenal cancer patients to verify whether ILC3, IL17A and DUOX2 or DUOXA2 (and others) are increased.

Due to intensive endoscopic surveillance and, when necessary, prophylactic duodenectomy for advanced duodenal polyposis that cannot be safely managed endoscopically, the incidence of duodenal carcinoma in FAP patients is fortunately rare, even in a large center like ours. As a result, for the purposes of this study, only duodenal carcinoma from one FAP patient for immunohistochemistry (IHC) was available. Nevertheless, we were able to confirm an increase in IL-17A(+)CD3(-) cells in FAP-associated duodenal carcinoma compared to tumor margins and non-FAP controls (Fig. 3A). Additionally, we observed a high number of DUOX2(+) epithelial cells in a representative staining of DUOX2 and IL-17A in duodenal carcinoma tissue from FAP patients (Supplementary Fig. 8E). Lines #432-434 and #580-582 deal with the changes made to the manuscript.

Additionally, we were able to analyze this sample by flow cytometry. In this specimen, NKp44(-) ILC3 cells were the predominant ILC population, further supporting the relevance of this subset in FAP-associated duodenal carcinoma (Figure for Review only 5A).

These findings strengthen our hypothesis that IL-17A(+)ILC3's play a relevant role in regulating DUOX2 and potentially contributing to carcinogenesis in the duodenum of FAP patients, likely driven by the increased presence of NKp44(-) ILC3 cells.

In vitro, it would be informative to show whether NKp44- (and not NKp44+) ILC3, in an IL17A dependent manner are able to convert human colonic adenoma cells (such as FPCK-1-1 cell line) into adenocarcinoma cells. Moreover, CRISPR/Cas9 technique is now well established, and the Authors should use it to KO/KD IL-17A receptor.

Thank you for your suggestion regarding the use of colonic adenoma cells and CRISPR/Cas9 technology to further explore the role of NKp44(-) ILC3 and IL-17A. However, we would like to emphasize that this line of investigation is not directly relevant to our current study. First, we have already demonstrated that NKp44(-) cells represent a smaller subpopulation of ILC3 in the colon compared to the duodenum, and these cells are not increased in colonic adenomas. Instead, ILC1s are more frequently found in colonic adenomas, suggesting that NKp44(-) ILC3s play a minor role in the context of colonic adenoma development. Second, the focus of our work is on the development of duodenal adenomas in FAP patients, and we kindly ask for your understanding that these proposed experiments fall beyond the scope of our study and were therefore not performed.

Regarding the use of CRISPR/Cas9 to knock out the IL-17A receptor, we understand that this is a valuable technique for determining the specificity of the IL-17A effect. However, we have already demonstrated that blocking IL-17A with a specific antibody, and not IL-17F, can inhibit the ILC3 supernatant-induced upregulation of DUOX2/DUOXA2 (Supplementary Fig. 8A). Additionally, our experiments (Fig. 7C) show that IL-17A, but not other cytokines like IL-1 β or IL-23, leads to the upregulation of DUOX2/DUOXA2. Since IL-17F can also bind to

the IL-17RA/RC receptor complex, using CRISPR/Cas9 to knock out this receptor would likely not provide further specific insights beyond what has already been shown through our antibody blockade experiments.

We appreciate your thoughtful suggestions, and while these avenues could be explored in future research, we believe that the current scope of our study already provides clear evidence of the specific role of IL-17A in the regulation of DUOX2/DUOXA2 and its contribution to carcinogenesis in FAP-associated duodenal adenomas.

Minor:

Figure 1A: please change the filling colour of the body, since then in the graphs 1C, D... the non-FAP are white, while the FAP are yellow/brown.

Figure 1B: the first contour plot is not clear, so it is difficult to see from where the lymphocytes were gated. Please change it.

yes

We have adopted both suggestions and improved the graphics in Fig. 1.

The choice of the lineage markers is interesting. Please justify it. Lim et al., Urberg et al. Moreover, in the discussion, please add few sentences stating that it was not possible to analyse CD4+ ILC1, CD5+ mature ILC1 and immature ILCs, NKp80+ NK-like ILCs...

Our own preliminary work with liver CD5(+)/CD4(+)-ILC1s in long-term cultures on OP9-DL4 showed that they mainly developed into T cells (<https://doi.org/10.1016/j.celrep.2022.111937>; Fig. S6H or Figure for review only 5B).

To avoid contamination of our ILC populations with T cells or T cell precursors, we included both CD4 and CD5 in the lineage cocktail. This was necessary to ensure the accuracy of our ILC3 analyses, given the potential for such cells to overlap phenotypically with ILCs.

We recognize that this strategy excluded CD4(+) and/or CD5(+) ILC subsets from our analysis. These subsets include potentially relevant populations such as CD4(+) ILC1s, CD5(+) mature ILC1s, immature ILCs, and NKp80(+) NK-like ILCs, which are known or hypothesized to exist in the human gut. While these subsets have been studied mainly in peripheral blood and lung tissues, their roles in the intestine remain largely unexplored (<https://doi.org/10.3389/fimmu.2017.01047>; doi: 10.3389/fimmu.2021.752104). By focusing on CD4(-) and CD5(-) ILCs, particularly ILC3s, our aim was to provide a more targeted analysis of ILC3 function in the duodenal microenvironment. This limitation has been addressed in the discussion (lines #701-703), where we acknowledge that CD4(+) and CD5(+) ILCs could not be analyzed in this study. This addition clarifies the scope of our analysis and provides a more comprehensive understanding of the ILC populations examined.

Supp Figure 1: I do not understand why ILC1 and ILC3 are negative for CD127 if they were gated as positive (according to the main Figure 1).

Thank you for your observation regarding the CD127 expression in Supplementary Fig. 1. We understand that the initial figure may have created the impression that ILC1 and ILC3 cells were CD127-negative. However, this was due to the absence of a clearly marked cut-off line (Supplementary Fig. 1A).

We have now revised the figure to include the cut-off line based on fluorescence minus one (FMO) controls, which clearly defines the CD127(+) ILC population. As you can see in the updated Supplementary Fig. 1A, the ILC1 and ILC3 populations are indeed CD127-positive, consistent with the gating strategy used in the main Fig. 1. We hope this clarifies the issue, and we appreciate your feedback.

Line 247: ILC3 number should be changed with frequency.

We have replaced this word with “frequency” (line #387). We are grateful for this suggestion.

Supp Figure 3B: I do not understand what the histograms on the right are quantifying, since on the left and from what the Authors claim is pretty clear that the frequency of colon NKp44+ ILC3 is increased. I am also misled by the colour code that is the one of the duodenum adenomas. The same colour code problem is present in figure 3 where the Authors are quantifying cells from the colon.

The histogram on the left side of Supplementary Fig. 3B represents the surface expression of NKp44 on duodenal versus colonic ILC3s, while the right side compares the frequency of NKp44(+) ILC3s in non-FAP colon tissue, normal FAP colon tissue, and FAP colon adenomas. However, based on your feedback, we recognize that the color coding and layout may have caused confusion.

In response to your comments, we have removed the right panel to simplify the figure and ensure clarity. The revised figure now focuses on the main points of interest, with Fig. 4A-C showing the distribution of ILCs among lymphocytes, the proportion of ILC3s among total lymphocytes, and the breakdown of ILC1, ILC2, NKp44(-) ILC3, and NKp44(+) ILC3 within the CD127(+) ILC population. This, we believe, provides a clear and detailed comparison of the relevant populations in the colon.

Additionally, the histogram in Supplementary Fig. 5B offers a broader overview of NKp44 expression on ILC3s across different compartments, complementing the data presented in the main figures. We hope these changes address the concerns and provide a more intuitive presentation of the data. Thank you again for your valuable feedback.

Figure 3B: “with the proportion of ILC3 among the overall colonic ILC pool being reduced in FAP adenoma”. No statistical analysis is provided to conclude this. The same for Figure 1F. The Authors should put standard error and statistical analysis in the bars. Supplementary Figure 1, Supplementary Figure 3

We apologize for any confusion caused and have addressed this by creating Supplementary Fig. 5C, which highlights a significant difference between FAP and FAP adenoma for colonic ILC3. Additionally, Fig. 4C now includes NKp44(+) and NKp44(-) ILC3s, demonstrating that the described ILC3 reduction specifically pertains to NKp44(+) ILC3s. We have also included the statistical analysis and error bars to the graphs. Furthermore, the newly added “Resource Data” Table provides the raw data associated with each figure, including details of the statistical tests performed.

Were the correlations shown in Figure 4 obtained with pool data from FAP and FAP adenomas? Since there are many dots, I guess this is the case. So, what happens to the correlations if the Authors split the 2 groups (even if one is small)?

The correlations in Fig. 4 were obtained using pooled data from the non-FAP, FAP, and FAP adenoma groups. This was necessary because only a portion of samples from each group had both gene expression and IL-17A+ NKp44(-) ILC3 frequency data available, and analyzing them separately would have resulted in sample sizes too small for robust statistical analysis.

The significant correlation is largely driven by the low expression of cytokines/Notch ligands and IL-17A+ NKp44(-) ILC3 frequency in the control group, while both parameters are elevated in the two FAP groups. To clarify this for readers and reviewers, we have incorporated a color-coded scheme for the three groups in the correlation graphs for IL1 β , IL23, Duox, and DuoxA2 (Fig. 5A-D).

Within the FAP and FAP adenoma groups, we observe a positive trend, but this correlation does not reach statistical significance (Figure for Review only 6A).

Despite the correlations with DLL1, the OP9DL1 to not induce changes in ILC3. The Authors should comment on this. During the revision of this manuscript, we re-evaluated all data and identified a minor error in the DLL1 dataset, where individual ILC3 data points had been incorrectly assigned to the corresponding DLL1 levels. After correcting this, the previously reported significant correlation between DLL1 transcript levels and IL-17A(+) NKp44(-) ILC3 frequency was no longer statistically significant (p-value = 0.06). We apologize for this oversight. Nevertheless, the correlation remains close to the significance threshold, and the trend observed in the original data persists.

This observation aligns with the broader context of our findings. While our *in vitro* experiments demonstrate that DLL1 does not directly influence IL-17A production by NKp44(-) ILC3s, the increased DLL1 transcript levels in FAP compared to non-FAP tissues remain significant. This increase is likely explained by the co-expression or spatial proximity of DLL1 and DLL4 within the tissue microenvironment, as evidenced by our MELC images (Fig. 5E and Supplementary Fig. 6A) and supported by a positive correlation between DLL1 and DLL4 levels (Figure for Review only 6B). This co-expression may underlie the observed association between DLL1 expression and IL-17A(+) NKp44(-) ILC3 frequency, though it does not imply a direct functional relationship.

Our data suggest that DLL4, rather than DLL1, plays a more critical role in inducing IL-17A production by NKp44(-) ILC3s, as demonstrated by our OP9-DL4 experiments. These findings have been clarified in the revised manuscript (lines #484-492), and the discussion has been expanded to emphasize the functional role of DLL4, IL-23, and IL-1 β in regulating IL-17A production (lines #656-663).

Reviewer #4 (Remarks to the Author): with expertise in FAP, intestinal oncology

The manuscript entitled 'IL-17A-Producing NKp44-ILC3s May Promote Duodenal Adenoma Formation in Familial Adenomatous Polyposis' by Kaiser and colleagues identifies an ILC population that is specifically enriched in the duodenum of FAP patients and is further increased in adenomatous duodenal samples of FAP patients. This particular population, NKp44-ILC3s, displays elevated IL-17A expression specifically in duodenal adenoma samples while not observed in colonic adenomas. The authors suggest that this population is involved in duodenal adenoma formation. Although the authors indeed demonstrate the presence of this particular population in duodenal adenomas, they do not provide any evidence for the involvement of this population in duodenal tumour formation. Due to complete lack of mechanistic insight this manuscript should not be eligible for Nature Communication in the current form. The following concerns were raised:

Major concerns
- No spatial validation of their findings. The entire manuscript is based on observations made using flow cytometry. The authors should demonstrate the presence of these specific populations in tissue samples. It currently remains unclear where these ILC3s are located in respect to the normal epithelium/adenomas, e.g. the mucosa/submucosa, near the crypt base/villus, top/base of the adenomas? In addition, by just using digested tissue biopsies it remains unclear what cells are being analysed by flow cytometry. For example, adenomatous tissue might also contain normal epithelium, and vice versa, normal epithelium might contain cells ta. The lack of spatial validation hampers the interpretation of the data and therefore prevents the identification of the mechanism by which IL-17A expressing cells might be involved in adenoma formation.

Thank you for raising this important point. We have addressed it by incorporating additional spatial validation into our study, which strengthens our findings and provides a clearer understanding of the role of NKp44(-)ILC3s in adenoma formation. These new analyses clarify the localization of these cells within the tissue and provide further evidence supporting their involvement in adenoma development.

To address concerns about tissue context, we used multi-epitope ligand cartography (MELC) microscopy to assess the spatial distribution of NKp44(-)ILC3s. These cells were predominantly located in the epithelial compartment, with fewer cells found in the lamina propria (Fig. 1F, Supplementary Fig. 3, Fig. 5E and Supplementary Fig. 6A). Furthermore, our flow cytometry data showed that IL-17A(+)NKp44(-) ILC3s in adenomas expressed high levels of epithelial retention markers, such as α E β 7 integrin (Fig. 3C and 3D, Supplementary Fig. 4B and 4C), indicating their retention within the epithelial tissue (lines #419-450).

In addition, our MELC analysis revealed that NKp44(-) ILC3s are found adjacent to CD31(+) endothelial cells expressing DLL4 (Fig. 5E and Supplementary Fig. 6A). This proximity suggests that endothelial signals, such as those of DLL4, may contribute to the accumulation and retention of these cells in the adenomas. To complement the MELC findings, we performed immunohistochemistry (IHC) to detect IL-17A, CD3 and nuclei and confirmed the presence of IL-17A(+) CD3(-) cells in tissue sections (Fig. 3A).

Regarding concerns about potential contamination between adenomatous and normal tissue, our results strongly suggest that the accumulation of IL-17A(+) NKp44(-) ILC3s is not due to contamination, but rather reflects true changes in the tissue. Although we cannot completely exclude the possibility that some of the areas analysed had already lost the second APC allele or contained undetectable microadenomas, the large number of samples, the consistency of our IHC/IF data and the clear spatial localisation of these cells make contamination an unlikely explanation. Instead, we propose that the accumulation of IL-17A(+) NKp44(-) ILC3s begins early in the normal mucosa and contributes to the changes in the microenvironment that facilitate adenoma formation. This accumulation becomes more pronounced within adenomas, as supported by both IHC/IF data and our flow cytometry analysis. This interpretation is further supported by our transcriptome data, which reveal the upregulation of tumor-associated genes such as CDH3, CEMIP, and DUOX2 in normal FAP tissue, highlighting their potential roles in tumorigenesis (Fig. 6D).

Finally, pathological assessment allowed us to distinguish between the central adenoma regions and the tissue margins based on markers such as the absence of goblet cells and increased nuclear density (Fig.3A and 8E). Our IHC data clearly show that IL-17A(+) CD3(-) cells are more abundant in FAP adenomas than in tissue margins, normal FAP tissue and non-FAP samples. This localization pattern further supports the hypothesis that NKp44(-) ILC3s play an active role in the adenoma microenvironment rather than being a mere epiphenomenon.

In conclusion, the spatial validation provided by these additional experiments reinforces our interpretation that NKp44(-) ILC3s accumulate specifically in adenomas and are likely to contribute to adenoma formation, rather than simply reflecting contamination or non-specific accumulation in the tissue.

- In line with the previous point, the identification of upregulated genes in FAP adenomas has not been validated spatially using tissue sections. Are these genes upregulated in FAP adenomas derived from the epithelial compartment, and can they be pinpointed specifically to adenomatous regions or the normal epithelium surrounding it? In addition, it remains unclear why FAP adenomas are compared to non-FAP control and not FAP controls in figure 5b to identify markers specifically upregulated in FAP adenomas. As validation of the bulk RNAseq data the authors

perform qPCR analysis, which raises the same concerns regarding biopsy digestion as the RNAseq and flow cytometry analysis.

Thank you very much for your valuable feedback. We acknowledge the concerns regarding the spatial validation of the upregulated genes identified in FAP adenomas. We have addressed these concerns by further supporting our hypothesis with protein level data, focusing in particular on DUOX2 expression, which corroborates our transcriptomic findings. As shown in Fig. 8D, immunofluorescence microscopy confirmed that DUOX2 is specifically upregulated in EpCAM(+) epithelial cells within FAP adenoma biopsies, indicating that the epithelial compartment is indeed responsible for the observed increase in DUOX2 expression. While it is difficult to distinguish between adenoma and adenoma margin in these images, Fig. 8E provides further clarity. The IHC staining clearly shows that DUOX2(+) epithelial cells are more frequently localized in the central adenomatous tissue compared to the margin. However, both the adenoma margin and even normal FAP tissue show higher DUOX2 expression than non-FAP controls, confirming that these processes extend beyond the adenoma core.

Fig. 8E also shows the close proximity of IL-17A(+) cells to DUOX2(+) cells. This supports our hypothesis that IL-17A(+) NKp44(-) ILC3s are likely to contribute to duodenal adenoma development through localized immune regulation. The spatial validation of DUOX2 expression in the epithelial compartment, combined with the close relationship with IL-17A(+) cells, strengthens the idea that this pathway plays a role in adenoma formation.

Regarding the comparison between FAP adenomas and non-FAP control tissue in old Fig. 5B, this was done to highlight the most significant transcriptomic differences. However, we agree that the inclusion of data from normal FAP tissue provides important additional information. To address this, we have extended our analysis, which is now included in the revised manuscript (Fig. 6B-E and Supplementary Data 2).

Despite the absence of notable gene alterations between normal FAP tissue and non-FAP controls in bulk-seq analysis, a greater number of DEGs were identified when comparing FAP adenomas to non-FAP tissue than when comparing them to normal FAP tissue. This may suggest that normal FAP tissue is beginning to undergo changes that could possibly be moving toward adenoma formation. Supporting this, qPCR analysis demonstrated upregulation of tumour-associated genes like *CEMIP*, *CDH3*, and *DUOX2* (Fig. 6B, 6C and 6D, Supplementary Data 2; Results section: lines #516-523 and #533-535; Discussion: lines #687-694).

In conclusion, the spatial validation by IHC and MELC, together with the extended transcriptomic analysis, strongly supports our conclusions. The upregulation of DUOX2, particularly in the epithelial compartment of adenomas, and its close association with IL-17A(+) NKp44(-) ILC3s, highlights the importance of this axis in the progression of duodenal adenomas in FAP patients.

- The authors suggest that ILC3s are involved in duodenal adenoma formation, however, by using tissues biopsies from established adenomas it remains unclear whether they have an active role in adenoma formation, or are just recruited to/enriched in established adenomas. If it has a role in adenoma formation, does IL17A addition to duodenal organoids from FAP patients drive oncogenic transformation? E.g. does it result in increased loss of APC in their *in vitro* cultures? Does it lead to morphological changes in the organoids? Does the cell type composition of the organoids change? Can they grow in the absence of Noggin and Wnt ligands? Again, no spatial/visual information regarding the organoid cultures is provided.

Our initial investigations identified an enrichment of IL-17(+) ILC3s in macroscopically normal mucosa of FAP patients (Fig. 2B; IL-17A(+)CD3(-) detected via IHC in Fig. 3A), suggesting a potential role for these cells in the early stages of adenoma formation. Additionally, we demonstrated that IL-17A (transcript levels shown in Fig. 7D, protein expression in Fig. 8F) and supernatants from NKp44(-) ILC3s (transcript levels shown in Fig. 8C and Supplementary Fig. 8A) upregulated DUOX2/DUOX2 expression in organoids, further supporting this hypothesis (Results section: lines #558-568, #580-584; Discussion: lines #635-638, #672-676, #687-694).

Following the reviewer's suggestion, we conducted additional experiments to further investigate the effects of IL-17A on duodenal organoids (Fig. 8F-I; Supplementary Fig. 8C-D). While IL-17A did not affect organoid proliferation (Supplementary Fig. 8C-D), it did result in the upregulation of DUOX2 and increased oxidative stress, as shown by the DC-FDA assay (Fig. 8F and 8I, Supplementary Fig. 8B). This oxidative stress likely caused elevated DNA damage, evidenced by γ H2AX staining (Fig. 8G-H, Supplementary Fig. 8B). Since oxidative DNA damage is a known driver of adenoma and cancer development, these findings suggest that IL-17A-producing NKp44(-) ILC3s contribute to FAP-associated adenoma and tumor formation through mechanisms involving oxidative stress. Additionally, spatial data of the organoids are presented in Fig. 8F-G, further illustrating these effects.

In terms of APC expression, we observed a slight but not significant reduction in APC levels in adenomatous tissue (Figure for Review only 7, q-value=0.097, non-FAP vs FAP adenoma). However, IL-17A treatment *in vitro* did not result in a decrease in APC gene expression in duodenal organoids (Supplementary Data 2, FAP organoid stimulation, adj.p-value=0.97; non-FAP organoid stimulation, adj.p-value=0.99). While this suggests that IL-17A does not directly induce APC loss, it is still possible that oxidative stress caused by IL-17A may promote other oncogenic mutations. Sustained oxidative stress can lead to persistent damage to both nuclear and mitochondrial DNA, potentially driving tumor progression through various pathways.

Furthermore, previous studies have demonstrated that loss of APC impairs normal regulation of reactive oxygen species (ROS), leading to oxidative stress and DNA damage. Therefore, we hypothesize that the combination of disrupted APC function and IL-17A signaling may accelerate tumor development in FAP patients by amplifying oxidative stress and its deleterious effects. This

hypothesis has been discussed in the manuscript, with supporting literature (DOI: 10.1016/j.stem.2013.04.006, lines #687-694).

- In line with the abovementioned point, is this population also enriched in sporadic duodenal adenomas or is it FAP specific? And can it be detected in duodenal carcinomas (FAP or sporadic)? Could it have a role in tumour progression instead of adenoma formation?

Unfortunately, we did not have access to samples of sporadic duodenal adenomas, as they are rare and no established screening programs exist to detect them in the general population.

However, we did examine a FAP-associated duodenal carcinoma from one patient and found an increased number of IL-17A(+) CD3(-) cells in these samples (Fig. 3A). This suggests that these cells may also play a role in tumor formation and progression. While it is conceivable that they contribute to later stages of tumorigenesis, our data show that these molecular and immune changes are already present in non-adenomatous tissue. Therefore, it is likely that early phases of adenoma formation are influenced by these cells, further underscoring their potential importance in FAP-related tumor development (Discussion: lines #687-694).

Minor comments:

- The y-axis labels between the duodenal samples (figure 1) and the colon samples (figure 3) are different to indicate the same population, this is confusing.

Thank you

According to this suggestion, we have adjusted the y-axis labeling to ensure consistency between Fig. 1 and Fig. 3, which now uniformly represent the same population.

- Some figures do not contain statistical (or n.s.) information, e.g. fig 2e,g,h or 3c.

Supplementary Figure 1 and 3

We appreciate the reviewer's observation regarding the missing statistical information. In response, we have now added the relevant significance levels for all group comparisons mentioned. However, for Supplementary Fig. 4, which includes six groups, we have decided not to display "n.s." to prevent the graph from becoming too cluttered and difficult to interpret. We believe this approach maintains clarity while still conveying the necessary information. Nevertheless, the complete raw data set has been collated and presented in the 'Resource data' table, accompanied by the statistical tests applied to each figure.

Reviewer #4 (Remarks to the Author):

In the revised version of their manuscript, the authors have addressed almost all my concerns. Particularly the addition of spatial information enhances the understanding of their work. However, there is one critical point that was already raised in my previous comments that remains unanswered, which is the actual proof that ILC3-produced IL17A promotes duodenal adenoma formation (as the title states).

In their revised manuscript, the authors demonstrate that IL17A induces DUOX expression, ROS production and DNA damage in normal duodenal organoids from FAP patients. Although such ROS-high/damaged/inflammatory environment most likely could contribute to the development of many types of cancers it is unclear how this 'promotes' duodenal adenoma development. As the authors are most likely aware of, the initiation and development of adenomas in FAP is driven by loss of the second APC allele, causing hyperactivation of the Wnt pathway and resulting in increased proliferation and changes in differentiation patterns. The authors reveal that there is no difference in proliferation/differentiation in their duodenal organoids, and APC-loss has not been investigated in their manuscript (which can e.g. be demonstrated by culturing organoids in the absence of growth factors such as Wnt3a and Rspodin). Of note: APC gene expression is not a reliable method for determining APC-loss of function as this expression is highly dependent on the specific mutation in the APC gene. Instead assessment of Wnt target gene upregulation would be a better indication.

Given that the functional implications of ILC3-produced IL17A on APC loss and thus on adenoma initiation cannot be demonstrated in the current work, I would suggest to rephrase the passages in the manuscript indicating that 'it promotes adenoma formation' and instead discuss that their results indicate that these ILC3s could contribute to a local environment that makes the epithelium more submissive for oncogenic transformation (or something similar).

Minor:

- It would be helpful if the authors could add to their figures whether the data is of duodenal or colon origin (perhaps as a small sub-figure header or something). I see this is already done in some figures but not all, and this would definitely contribute to the readability/interpretation of the results as some of the figures throughout the manuscript look quite alike.
- Figure 3c the z-score legend is left completely blue (instead of Red-Yellow-Blue).
- Figure 5a, I think the y-axis of the correlation plot should be IL1B not IL23?
- Sup. Figure 2a y-axis IL-8 instead of 'IL-8'

Response

We appreciate the critical feedback from **Reviewer #4** and fully understand the concerns raised. In response, we have implemented several changes throughout the manuscript:

- **Terminology Revision:** We have removed the term “adenoma formation” and replaced it with more precise language, such as “**oncogenic transformation,**” “**malignant transformation,**” or “**tumorigenic progression.**” This change reflects our intent to suggest that ILC3-produced IL17A may contribute to a local environment that predisposes the epithelium to oncogenic changes.

- **Figure Annotations:** To improve clarity and data interpretation, we have added annotations to our figures indicating the tissue origin (duodenal or colonic) where it was missing. This enhancement ensures that all figures now clearly convey whether the data are derived from duodenal or colon tissue.

- **Figure Corrections:**

- In **Figure 3c**, we have corrected the z-score legend to display the appropriate Red-Yellow-Blue gradient.

- In **Figure 5a**, we have updated the y-axis label from IL23 to ***IL1B***.

- In **Supplementary Figure 2a**, the typographical error in the y-axis label has been fixed, changing “IIL-8” to ***IL-8***.

These modifications are now implemented in our revised manuscript. We believe these changes address the reviewer’s concerns and significantly enhance the clarity and accuracy of our work.